



# Development of Korean Air Quality Prediction System version 1 (KAQPS v1): an operational air quality prediction system with focuses on practical issues

**Kyunghwa Lee[1,2], Jinhyeok Yu[2], Sojin Lee[3], Mieun Park[4,5], Hun Hong[2], Soon Young Park[2], Myungje Choi[6], Jhoon Kim[6], Younha Kim[7], Jung-Hun Woo[7], Sang-Woo Kim[8] and Chul H. Song[2*]**

1. Environmental Satellite Center, Climate and Air Quality Research Department, National Institute of Environmental Research (NIER), Incheon, Republic of Korea
2. School of Earth Sciences and Environmental Engineering, Gwangju Institute of Science and Technology (GIST), Gwangju, Republic of Korea
3. Department of Earth and Atmospheric Sciences, University of Houston, Texas, USA
4. Air Quality Forecasting Center, Climate and Air Quality Research Department, National Institute of Environmental Research (NIER), Incheon, Republic of Korea
5. Environmental Meteorology Research Division, National Institute of Meteorological Sciences (NIMS), Jeju, Republic of Korea
6. Department of Atmospheric Sciences, Yonsei University, Seoul, Republic of Korea
7. Department of Advanced Technology Fusion, Konkuk University, Seoul, Republic of Korea
8. School of Earth and Environmental Sciences, Seoul National University, Seoul, Republic of Korea

**Short title:** Operational air quality prediction system in Korea

**Corresponding author: Chul H. Song (chsong@gist.ac.kr)**



**Abstract**

For the purpose of providing reliable and robust air quality predictions, an
operational air quality prediction system was developed for the main air quality criteria
species in South Korea ($PM_{10}$, $PM_{2.5}$, CO, $O_3$, and $SO_2$). The main caveat of the system is to
prepare the initial conditions (ICs) of the Community Multi-scale Air Quality (CMAQ)
model simulations using observations from the Geostationary Ocean Color Imager (GOCI)
and ground-based monitoring networks in northeast Asia. The performance of the air quality
prediction system was evaluated during the Korea-United States Air Quality Study (KORUS-
AQ) campaign period (1 May–12 June 2016). Data assimilation (DA) of optimal
interpolation (OI) with Kalman filter was used in this study. One major advantage of the
system is that it can predict not only particulate matter (PM) concentrations but also PM
chemical composition including five main constituents: sulfate ($SO_4^{2-}$), nitrate ($NO_3^-$),
ammonium ($NH_4^+$), organic aerosols (OAs), and elemental carbon (EC). In addition, it is also
capable of predicting the concentrations of gaseous pollutants (CO, $O_3$ and $SO_2$). In this sense,
this new operational air quality prediction system is comprehensive. The results with the ICs
(DA RUN) were compared with those of the CMAQ simulations without ICs (BASE RUN).
For almost all of the species, the application of ICs led to improved performance in terms of
correlation, errors, and biases over the entire campaign period. The DA RUN agreed
reasonably well with the observations for $PM_{10}$ (IOA = 0.60; MB = -13.54) and $PM_{2.5}$ (IOA =
0.71; MB = -2.43) as compared to the BASE RUN for $PM_{10}$ (IOA = 0.51; MB = -27.18) and
$PM_{2.5}$ (IOA = 0.67; MB = -9.9). A significant improvement was also found with the DA RUN
in terms of bias. For example, for CO, the MB of -0.27 (BASE RUN) was greatly enhanced
to -0.036 (DA RUN). In the cases of $O_3$ and $SO_2$, the DA RUN also showed better



performance than the BASE RUN. Further, several more practical issues frequently
encountered in the operational air quality prediction system were also discussed. In order to
attain more accurate ozone predictions, the DA of $NO_2$ mixing ratios should be implemented
with careful consideration of the measurement artifacts (i.e., inclusion of alkyl nitrates, $HNO_3$,
and PANs in the ground-observed $NO_2$ mixing ratios). It was also discussed that, in order to
ensure accurate nocturnal predictions of the concentrations of the ambient species, accurate
predictions of the mixing layer heights (MLH) should be achieved from the meteorological
modeling. Several advantages of the current air quality prediction system, such as its non-
static free parameter scheme, dust episode prediction, and possible multiple implementations
of DA prior to actual predictions, were also discussed. These configurations are all possible
because the current DA system is not computationally expensive. In the ongoing and future
works, more advanced DA techniques such as the three-dimensional variational (3DVAR)
method and ensemble Kalman filter (EnK) are being tested and will be introduced to the
Korean operational air quality forecasting system.

**Keywords:** Air quality prediction; Particulate matter (PM); Geostationary satellite sensor
(GOCI); Air Korea; Data assimilation (DA); Dust episode predictions; $NO_2$ measurement
artifacts





**1. Introduction**

Air quality has long been considered an important issue in climate change, visibility,

and public health, and it is strongly dependent upon meteorological conditions, emissions,
and the transport of air pollutants. Air pollutants typically consist of atmospheric particles and
gases such as particulate matter (PM), carbon monoxide (CO), ozone ($O_3$), nitrogen dioxide
($NO_2$), and sulfur dioxide ($SO_2$). These aerosols and gases play important roles in
anthropogenic climate forcing both directly (Bellouin et al., 2005; Carmichael et al., 2009;
IPCC, 2013; Scott et al., 2014) and indirectly (Bréon et al., 2002; IPCC, 2013; Penner et al.,
2004; Scott et al., 2014) in influencing the global radiation budget. Among the various air
pollutants, PM and surface $O_3$ are the most notorious health threats, as has been stated by
several previous studies (e.g. Carmichael et al., 2009; Dehghani et al., 2017; Khaniabadi et al.,

2017).

With the stated importance of atmospheric aerosols and gases, considerable research

efforts have been made to monitor and quantify their amounts in the atmosphere through
satellite-, airborne-, and ground-based observations as well as chemistry-transport model
(CTM) simulations. In South Korea, the Korean Ministry of the Environment (KMoE)
provides real-time chemical concentrations as measured by ground-based observations for six
criteria air pollutants ($PM_{10}$, $PM_{2.5}$, $O_3$, CO, $SO_2$, and $NO_2$) at the Air Korea website
(https://www.airkorea.or.kr). In addition, the National Institute of Environmental Research
(NIER) of South Korea provides air quality (chemical weather) predictions using multiple
CTM simulations. Air quality predictions are another crucial element for protecting public
health through the forecasting of high air pollution episodes in advance and alerting citizens
about these high episodes. In this context, reliable and robust chemical weather forecasts are





necessary to avoid any confusion caused by poor predictions given by CTM simulations.

Although there are various datasets representing air quality, limitations remain in the

observations and model outputs. Specifically, observation data are, in general, known to be
more accurate than model outputs, but they have spatial and temporal limitations. Unlike
observation data, models can provide meteorological and chemical information without any
spatial and temporal data discontinuity, but they do have an issue of inaccuracy. The major
causes of uncertainty in the results of CTM simulations are introduced from imperfect
emissions, meteorological fields, initial conditions (ICs), and physical and chemical
parameterizations in the models (Carmichael et al., 2008). In order to minimize the
limitations and maximize the advantages of observation data and model outputs, there have
been numerous attempts to provide accurate and spatially- as well as temporally- continuous
information on chemical composition in the atmosphere by integrating observation data with
model outputs via data assimilation (DA) techniques.

Although the Korean operational numerical weather prediction (NWP) carried out by

the Korea Meteorological Administration (KMA) employs various DA techniques, almost no
previous efforts have been made to develop a chemical weather prediction system with DA in
South Korea. Therefore, in the present study, an operational chemical weather prediction
system named as Korean Air Quality Prediction System version 1 (KAQPS v1) was
developed by preparing ICs via DA for the Community Multi-scale Air Quality (CMAQ)
model (Byun and Schere, 2006; Byun and Ching, 1999) using satellite- and ground-based
observations for particulate matter (PM) and atmospheric gases such as CO, $O_3$, and $SO_2$. The
performances of the system were then demonstrated during the period of the Korea-United
States Air Quality Study (KORUS-AQ) campaign (1 May – 12 June 2016) in South Korea.



In this study, the optimal interpolation (OI) method with the Kalman filter was
applied in order to develop an operational air quality prediction system, since this method is
still useful and viable in terms of computational cost and performance. The performance of
the method is almost comparable to that of the three-dimensional variational (3DVAR)
method, as shown in Tang et al. (2017). More complex and advanced DA techniques are
currently being and will continue to be applied to current air quality prediction systems.
These works are now in progress.
In addition, this manuscript also discusses several practical issues frequently
encountered in the operational air quality predictions such as: i) DA of $NO_2$ mixing ratios for
accurate ozone prediction with a careful consideration of measurement artifacts; ii) the issue
of the nocturnal mixing layer height (MLH) for nocturnal predictions; iii) predictions of dust
episodes; iv) the use of non-static free parameters; and v) the influences of multiple
implementations of the DA before the actual predictions.
The details of the datasets and methodology used in this study are described in Sect.
2. The results of the developed operational chemical weather prediction system are discussed
in Sect. 3, and then a summary and conclusions are given in Sect. 4.

**2. Methodology**

The operational air quality prediction system was developed using the CMAQ model
along with meteorological inputs provided by the Weather Research and Forecasting (WRF)
model (Skamarock et al., 2008). The ICs for the CMAQ model simulations were prepared via
the DA method using satellite-retrieved and ground-based observations. The performances of
the developed prediction system were evaluated using ground in-situ data. The models, data,
and DA technique are described in detail in the following sections.



### 2.1 Meteorological and chemistry-transport modeling

#### 2.1.1 WRF model simulations

The WRF model has been developed for providing mesoscale numerical weather prediction (NWP). It has also been used to provide meteorological input fields for CTM simulations (Appel et al., 2010; Chemel et al., 2010; Foley et al., 2010; Lee et al., 2016; Park et al., 2014). In this study, WRF v3.8.1 with the Advanced Research WRF (ARW) dynamic core was applied to prepare the meteorological inputs for the CMAQ model simulations. The National Centers for Environmental Prediction Final Analysis data (NCEP FNL) were chosen for the ICs and boundary conditions (BCs) for the WRF simulations. In order to minimize meteorological field error, the objective analysis (OBSGRID) nudging was conducted using the NCEP Automated Data Processing (ADP) global upper-air/surface observational weather data. The model domain for the WRF simulations covers Northeast Asia with a horizontal resolution of $15 \times 15$ km$^2$, having a total of 223 latitudinal and 292 longitudinal grid cells. The size of the WRF domain is slightly larger than that of the CMAQ domain, as shown in Fig. 1. The meteorological data also have 27 vertical layers from the surface (1000 hPa) to 50 hPa.

#### 2.1.2 CMAQ model simulations

The CMAQ v5.1 model was used to estimate the concentrations of the atmospheric chemical species over the domain, as shown in Fig. 1. The CMAQ domain has 204 latitudinal and 273 longitudinal grid cells in total, and also has a $15 \times 15$ km$^2$ horizontal resolution and 27 sigma vertical layers. For anthropogenic emissions, KORUS v1.0 emissions (Woo et al.,





2012) were used. The KORUS v1.0 emissions cover almost all of Asia, and are based on
three emission inventories: the Comprehensive Regional Emissions inventory for
Atmospheric Transport Experiment (CREATE) for East Asia excluding Japan; the Model
Inter-Comparison Study for Asia (MICS-Asia) for Japan; and the Studies of Emissions and
Atmospheric Composition, Clouds and Climate Coupling by Regional Surveys (SEAC4RS)
for South and Southeast Asia.

Biogenic emissions were prepared by running the Model of Emissions of Gases and

Aerosols from Nature (MEGAN v2.1; Guenther et al., 2006, 2012) with a grid size identical
to that of the CMAQ model simulations. For the MEGAN simulations, the MODIS land
cover data (Friedl et al., 2010) and improved leaf area index (LAI) based on MODIS datasets
(Yuan et al., 2011) were utilized. Pyrogenic emissions were obtained from the Fire Inventory
from NCAR (FINN; Wiedinmyer et al., 2006, 2011). The lateral BCs for the CMAQ model
simulations were prepared using the global model results of the Model for Ozone and Related
chemical Tracers version 4 (MOZART-4; Emmons et al., 2010) at every 6 hours. The
mapping and re-gridding of the MOZART-4 data were conducted by matching the CMAQ
grid information.

**2.2 Observation data**
**2.2.1. Satellite-based observations**

A Korean geostationary satellite of Communication, Ocean, and Meteorological

Satellite (COMS) was launched on 26 June in 2010 over the Korean Peninsula. The COMS is
a geostationary orbit satellite and it is stationed at an altitude of approximately 36,000 km at a
latitude of 36°N and a longitude of 128.2°E with a horizontal coverage of $2500 \times 2500$ km$^2$
(refer to Fig. 1). Among the three payloads of the COMS, Geostationary Ocean Color Image



(GOCI) is the first multi-channel ocean color sensor with visible and near infrared channels.
The GOCI instrument provides hourly spectral images with a spatial resolution of $500 \times 500$
$m^2$ from 00:30 to 07:30 Coordinated Universal Time (UTC) for eight spectral (6 visible and 2
near-infrared) channels at 412, 443, 490, 555, 660, 680, 745, and 865 nm.

The Yonsei aerosol retrieval (YAER) algorithm for the GOCI sensor was initially

developed by Lee et al. (2010) to retrieve the aerosol optical properties (AOPs) over ocean
areas, and was then improved by expanding to consider non-spherical aerosol optical
properties (Lee et al., 2012). Choi et al. (2016) further extended the algorithm for application
to land surfaces, and the algorithm was referred to as the GOCI YAER version 1 algorithm.
With the GOCI YAER algorithm, hourly Aerosol Optical Depths (AODs) at 550 nm were
produced over East Asia. Choi et al. (2016) compared the retrieved GOCI AODs with other
satellite-retrieved and ground-based observations, and found several errors in the cloud
masking and surface reflectances. These errors were corrected in the recently updated second
version of the GOCI YAER algorithm (Choi et al., 2018), which used the updated cloud
masking and more accurate surface reflectances. In this study, the most recent GOCI AOD
products from the GOCI YAER version 2 algorithm were used.

**2.2.2. Ground-based observations**

In addition to the satellite data, ground-based observations in South Korea and China

were also collected for use in the operational air quality prediction system for PM and gas-
phase pollutants. The orange, red, and blue dots in Fig. 1 represent the ground-based
observation sites in China, Air Korea, and super-site stations in South Korea, respectively.
These observations provide real-time concentrations of criteria species such as $PM_{10}$, $PM_{2.5}$,





CO, $O_3$, $SO_2$, and $NO_2$.
Throughout the period of the KORUS-AQ campaign, ground-based observation data
were obtained from 1514 stations in China, 264 Air Korea stations, and seven super-site
stations in South Korea. In this study, 80 % of the ground-based observations in China and
Air Korea stations in South Korea were randomly selected for use in the prediction system.
The other 20 % of the data and super-site observations were used to evaluate the
performances of the developed air quality prediction system.
In addition, AErosol RObotic NETwork (AERONET) AODs were used to conduct an
independent evaluation of the air quality prediction system. AERONET is a federated global
ground-based sun photometer network (Holben et al., 1998). Cloud-screened and quality-
assured level 2.0 AODs for the AERONET were used in this study.

**2.3 Operational air quality prediction system**
In the present study, the operational air quality prediction system was developed by
adjusting the ICs for the CMAQ model simulations based on DA with satellite-retrieved and
ground-measured observations. Two parallel WRF-CMAQ model runs were conducted. The
first experiment that involved adjusting ICs via DA is referred to as DA RUN (see Fig. 2). In
order to evaluate the prediction system, a second experiment, in which the ICs were
originated from the previous CMAQ model simulations without assimilations, was also
conducted. This CMAQ run is referred to as BASE RUN.

**2.3.1. AOD calculations**
CMAQ AODs are calculated by integrating the aerosol extinction coefficient ($\sigma_{ext}$)



using the following equation:

$$AOD(\lambda) = \int_0^z \sigma_{ext}(\lambda) \ dz \tag{1}$$


where z represents the vertical height; $\sigma_{ext}$ is defined as the sum of the absorption
coefficient ($\sigma_{abs}$) and the scattering coefficient ($\sigma_{sca}$); and $\sigma_{abs}$ and $\sigma_{sca}$ can be estimated
by Eqns (3) and (4), respectively, as shown below:

$$\sigma_{ext}(\lambda) = \sigma_{abs}(\lambda) + \sigma_{sca}(\lambda) \tag{2}$$

$$\sigma_{abs}(\lambda) \ [Mm^{-1}] = \sum_i^n \sum_j^m \{(1 - \omega_{ij}(\lambda)) \cdot \beta_{ij}(\lambda) \cdot f_{ij}(RH) \cdot [C]_{ij}\} \tag{3}$$
$$\sigma_{sca}(\lambda) \ [Mm^{-1}] = \sum_i^n \sum_j^m \{\omega_{ij}(\lambda) \cdot \beta_{ij}(\lambda) \cdot f_{ij}(RH) \cdot [C]_{ij}\} \tag{4}$$

where i and j denote the particulate species and size bin (or particle mode), respectively;
$\omega_{ij}(\lambda)$ is the single scattering albedo; $\beta_{ij}(\lambda)$ is the mass extinction efficiency (MEE) of
particulate species i for the size bin or particle mode j; $[C]_{ij}$ is the concentration of
particulate species including $(NH_4)_2SO_4$, $NH_4NO_3$, black carbon, organic aerosols (OA),
mineral dust, and sea-salt aerosols; RH is the relative humidity; and $f_{ij}(RH)$ is the
hygroscopic factor.
Here, the single scattering albedo ($\omega$) refers to the fraction (portion) of the scattering
over total extinction. In this work, $\sigma_{ext}$ was estimated using $\beta$ and f(RH), as suggested by
Chin et al. (2012). Park et al. (2014) and Lee et al. (2016) found that the values reported by
Chin et al. (2012) produced the best results in estimating AODs at 550 nm over East Asia.
The calculated AODs were used in the air quality prediction system in order to prepare the



ICs for the PM predictions.

**2.3.2. Data assimilation (DA)**

The ground-based observations, together with GOCI-derived AODs, were used to

prepare the ICs for the operational air quality predictions with the CMAQ model simulations.
In order to achieve this, the following steps were taken: (i) the CMAQ-calculated
concentrations of CO, $O_3$, and $SO_2$ were combined with the concentrations of CO, $O_3$, and
$SO_2$ obtained from ground-based observations in South Korea (Air Korea) and China; (ii) the
CMAQ-calculated AODs were assimilated with the GOCI AODs; (iii) the assimilated AODs
were converted into $PM_{10}$; (iv) the converted $PM_{10}$ was again assimilated at the surface in
South Korea and China; and (v) after the DA at the surface, the ratios of the assimilated
species concentrations to the original CMAQ-simulated concentrations were applied so as to
the adjust vertical profiles of the chemical species above the surface. In the operational
prediction system, the DA cycle is 24 hours and the assimilation takes place every day at
00:00 UTC (refer to Fig. 3).

The optimal interpolation (OI) method with the Kalman filter was chosen in the

operational air quality prediction system. The OI method was originally used for
meteorological applications (Lorenc, 1986), and has also been used in the assimilations for
trace gases (Khattatov et al., 1999, 2000; Lamarque et al., 1999; Levelt et al., 1998). Recently,
the OI technique has also been applied to aerosol fields (Collins et al., 2001; Yu et al., 2003;
Generoso et al., 2007; Adhikary et al., 2008; Carmichael et al., 2009; Chung et al., 2010; Park
et al., 2011; Tang et al., 2015, 2017).

Aerosol assimilation using the OI method was first applied by Collins et al. (2001) as





follows:

$$\tau'_m = \tau_m + \mathbf{K}(\tau_o - \mathbf{H}\tau_m) \tag{5}$$

$$\mathbf{K} = \mathbf{B}\mathbf{H}^T(\mathbf{H}\mathbf{B}\mathbf{H}^T + \mathbf{O})^{-1} \tag{6}$$

$$\mathbf{O} = [(f_o\tau_o)^2 + (\varepsilon_o)^2]\mathbf{I} \tag{7}$$

$$\mathbf{B}(d_x, d_z) = [(f_m\tau_m)^2 + (\varepsilon_m)^2]\exp\left[-\frac{d_x^2}{2l_{mx}^2}\right]\exp\left[-\frac{d_z^2}{2l_{mz}^2}\right] \tag{8}$$


where $\tau'_m$, $\tau_m$, and $\tau_o$ represent the assimilated products by the OI method, the modeled
values, and the observed values, respectively; $\mathbf{K}$ is the Kalman gain matrix; $\mathbf{H}$ is the
observation operator (or forward operator), which is an interpolator from model to
observation space; $\mathbf{B}$ and $\mathbf{O}$ are the background and observation error covariance matrices,
respectively; $(\cdot)^T$ denotes the transpose of a matrix; $f_o$ is the fractional error in the
observation-retrieved value; $\varepsilon_o$ is the minimum root mean square error in the observation-
retrieved values; $f_m$ is the fractional error in the model estimates; $\varepsilon_m$ is the minimum root
mean square error in the model estimates; $d_x$ is the horizontal distance between two model
grid points; $l_{mx}$ is the horizontal correlation length scale for the errors in the model; $d_z$ is
the vertical distance between two model grid points; and $l_{mz}$ is the vertical correlation
length scale for the errors in the model. In this work, the OI technique was applied for the DA
of atmospheric gaseous species as well as particulate species.

Six free parameters ($f_m$, $f_o$, $\varepsilon_m$, $\varepsilon_o$, $l_{mx}$, and $l_{mz}$) were used to calculate the error

covariance matrices of the observations and model, the mathematical formalisms of which
are described in Eq. (7) and (8), respectively. Several previous studies have used fixed values
for free parameters (Collins et al., 2001; Yu et al., 2003; Adhikary et al., 2008; Chung et al.,



2010). These runs are called "static" runs. In contrast to those previous studies, "non-static"
free parameters were applied in this study by minimizing the differences between the
assimilated values and observations via an iterative process at each assimilation time step.
This non-static free parameter scheme is possible due to the fact that the OI technique with
the Kalman filter is much less costly in terms of computation time than other DA techniques,
such as the 3-D or 4-D variational methods. This is another advantage of using the OI
technique in this system. It typically takes less than 20 minutes with a workstation
environment (dual Intel Xeon 2.40 GHz processor).

**2.3.3. Allocation of the assimilated $PM_{10}$ & $PM_{2.5}$ into particulate composition**

In the procedure of operational DA, $PM_{10}$ was assimilated in this study, because the

$PM_{10}$ data were more plentiful than $PM_{2.5}$. The assimilated $PM_{10}$ then needs to be allocated
into the PM composition for the CMAQ-model prediction runs. In order to achieve this, the
differences between the assimilated $PM_{10}$ and background $PM_{10}$ ($\Delta PM_{10}$) were first calculated.
Then, $\Delta PM_{2.5}$ was estimated using the ratios of $PM_{2.5}$ to $PM_{10}$ from the background CMAQ
model runs (i.e., $\Delta PM_{2.5}=\Delta PM_{10}\times PM_{2.5}/PM_{10}$). $\Delta PM_{2.5}$ was then allocated to the $PM_{2.5}$
composition according to the comparison between two $PM_{2.5}$ compositions observed at the
seven super-sites and simulated from the CMAQ model runs over South Korea. Both of the
compositions are shown in Fig. 4. In Fig. 4, "PM OTHERS" indicates the remaining
particulate matter species after excluding sulfate, nitrate, ammonium, organic aerosol (OA),
and elementary carbon (EC). The PM OTHERS occupies 25 % of the total $PM_{2.5}$ observed at
super-sites. The other fraction, $\Delta PM_{10}\times(1-PM_{2.5}/PM_{10})$, was also distributed into the coarse-
mode particles ($PM_{2.5-10}$) as crustal elements.





## 3. Results and discussions

The performances of the air quality prediction system were evaluated by comparing them with ground-based observations from the Air Korea network and super-site stations in South Korea. Several sensitivity analyses were also conducted in order to assess the influences of the DA time-intervals on the accuracy of the air quality prediction.

### 3.1. Evaluation of the air quality prediction system

#### 3.1.1. Time-series analysis

Figure 5 shows the time-series plots of $PM_{10}$, $PM_{2.5}$, CO, $O_3$, and $SO_2$ concentrations from the BASE RUN and the DA RUN. Here, the observation data (OBS) obtained from the Air Korea network were compared with the results of the two sets of the CMAQ model simulations, i.e., (1) BASE RUN and (2) DA RUN. As mentioned previously, 20% of the Air Korea observations used in the evaluation were randomly selected during the period of the KORUS-AQ campaign. The other 80 % of the Air Korea data were used in the DA at 00:00 UTC. For the forecast hours from 01:00 to 23:00 UTC, all of the ground observations (254 Air Korea and seven super-site stations) were used to evaluate the performances of the developed air quality prediction system. As shown in Fig. 5, we achieved some improvements in the prediction performances by applying the ICs to the CMAQ model simulations. The BASE RUN significantly under-predicted $PM_{10}$, $PM_{2.5}$, and CO while the DA RUN produced concentrations that were more consistent with the observations than those of the BASE RUN.

In case of CO, the observed CO mixing ratios were about three times higher than those from the BASE RUN. These large differences are well known, and have been attributed to the underestimated emissions of CO (Heald et al., 2004). However, when the DA was



applied, the predictions of the CO mixing ratios improved. Similarly, the performances of the
$PM_{10}$ and $PM_{2.5}$ predictions improved with the application of the DA. Unlike $PM_{10}$, $PM_{2.5}$,
and CO, the $O_3$ mixing ratios and its diurnal trends from both the BASE RUN and DA RUN
tend to be well-matched with the observations. By contrast, the poorest performances of the
BASE RUN and the DA RUN were shown for $SO_2$.
In addition, a dust event took place between 6 May and 8 May. This event is captured
by the DA RUN (check red peaks in Fig. 5(a) and (b)), while the BASE RUN cannot capture
this dust event. This demonstrates the capability of the current system to possibly predict dust
events in South Korea. In the DA RUN, dust information is provided into the CMAQ model
runs through both/either GOCI AOD and/or ground PM observations measured along the dust
plume tracks.
The effectiveness of the DA with prediction time was also analyzed by calculating
the aggregated average concentrations of atmospheric species (see Figs. 6, 7, and 9). Fig. 6
depicts the CMAQ-calculated average concentrations of $PM_{10}$, $PM_{2.5}$, CO, and $SO_2$ against
the Air Korea observations. Our air quality prediction system re-generated relatively well-
matched concentrations for $PM_{10}$, $PM_{2.5}$, and CO from the DA RUN.
Figure 7 shows the case of ozone. The ozone mixing ratios from the DA RUN in Fig.
7(a) were reasonably consistent with the observations at 00:00 UTC, but disagreed with those
between 04:00 and 09:00 UTC (13:00 and 18:00 KST), when solar insolation is the most
intense. This may be attributed to the chemical imbalances between ozone production and
ozone destruction (or titration). However, if CMAQ $NO_2$ was assimilated with ground-based
observations in South Korea (Air Korea) and China, the predicted ozone mixing ratios
became substantially closer to the observations, as shown in Fig. 7(b). This is clearly due to



the fact that $NO_x$ is an important precursor of ozone. In the prediction of the ozone mixing
ratios, both 1-hr peak ozone (around 15:00 KST) and 8-hr averaged ozone mixing ratios
(between 9:00 and 17:00 KST) are important. Fig. 7 clearly shows that the prediction
accuracies of both the ozone mixing ratios were improved after the DA of $NO_2$ mixing ratios.
Although the DA for $NO_2$ provided better ozone predictions, one should take caution
in using the $NO_2$ observations. The $NO_2$ mixing ratios measured at Air Korea sites are known
to be contaminated by other nitrogen gases such as nitric acid ($HNO_3$), peroxyacetyl nitrates
(PANs), and alkyl nitrates (ANs), since the Air Korea $NO_2$ mixing ratios are measured
through a chemiluminescent method with catalysts of gold or molybdenum oxide at high
temperatures. These are known to be "$NO_2$ measurement artifacts" (Jung et al., 2017), which
is one of the reasons that the DA of $NO_2$ was not shown in Fig. 6. The $NO_2$ mixing ratios are
corrected from the Air Korea $NO_2$ data, and are then used to prepare the ICs via the DA for
more accurate ozone and $NO_2$ predictions. Currently, such corrections of the observed $NO_2$
mixing ratios are being standardized with more sophisticated year-long $NO_2$ measurements.
After the corrections of the $NO_2$ measurement artifacts, more evolved schemes of ozone and
$NO_2$ predictions will be possible in the future. As shown in Fig. 7, about a 20% reduction
(average fraction of non-$NO_2$ mixing ratios in the observed $NO_2$ mixing ratios) was made for
these demonstration runs (Jung et al., 2017)**.**
Another practical issue is now discussed. Although the assimilation with the
observed $NO_2$ mixing ratios can enhance the accuracy of the predictions of the daytime ozone
mixing ratios, the nighttime ozone mixing ratios tend to be consistently over-predicted in the
aggregated plot of the ozone mixing ratios at the observation sites (see Fig. 7). This is
believed to be caused by underestimation of the mixing layer height (MLH). Figure 8 shows a





comparison between lidar-measured MLH (black dashed line) and WRF-calculated MLH
(with the option of the Yonsei University (YSU) planetary boundary layer scheme) (Hong et
al., 2006; see red line). As shown in Fig. 8, the nocturnal lidar-measured MLH is about two
times higher than the nocturnal WRF-calculated MLH as measured at a lidar site inside the
campus of Seoul National University (SNU) in Seoul. This is a common and well-defined
phenomenon in East Asia. Such underestimated MLH in the model tends to compress the
ozone molecules within the mixing layer during the nighttime, which leads to consistently
over-predicted nocturnal ozone mixing ratios.

Although the correct predictions of the daytime ozone mixing ratios are substantially

more important, it is also worth trying to achieve correct predictions of the nocturnal ozone
mixing ratios. Correct predictions of the nocturnal ozone mixing ratios strongly depend on
the correct estimation of the MLH. Currently, efforts are being made in two directions. First,
a modified MLH (or PBL) scheme in the meteorological model is currently being studied.
The other area of study is that the WRF-calculated MLH can be "bias-corrected" to match the
observed MLHs in the interface (MCIP) between a MET model (e.g., WRF) and a CTM
model (e.g., CMAQ). These efforts are now underway as well.

In this work, the aerosol composition (such as EC, OA, sulfate, nitrate, and

ammonium) was further compared with the composition observed at the super-sites shown in
Fig. 9. As shown in Fig. 9, agreement was observed between the DA RUN and observations
for all of the major PM constituents. Again, this is another strong capability of our system for
predicting not only particle mass, but also the "chemical composition" of particulate matters.

**3.1.2. Spatial distribution**





Figure 10 shows the spatial distributions of PM and chemical species throughout the
entire period of the KORUS-AQ campaign over the Seoul Metropolitan Area (SMA).
Noticeable improvements are observed to have been achieved in the spatial distributions by
applying the ICs into the CMAQ model simulations, particularly for $PM_{10}$ (Fig. 10a), $PM_{2.5}$
(Fig. 10b), and CO (Fig. 10c). As shown in Fig. 10, the under-predicted concentrations of
$PM_{10}$, $PM_{2.5}$, and CO were adjusted to concentrations closer to the observations. In case of
$SO_2$ (see Fig. 10d), the DA RUN produced better agreement with the observations than the
BASE RUN, but there were still under-predicted $SO_2$ concentrations over the northeastern
part of the SMA.
By contrast, relatively lower ozone mixing ratios from the DA RUN against the
BASE RUN were found in the southwestern part of the SMA (see Fig. 10e). Due to the
nonlinear relationship between $NO_x$ and $O_3$, high mixing ratios (or emissions) of $NO_x$ in the
SMA can lead to depletion of ozone. In these runs, the precursors of ozone such as $NO_x$ and
VOCs were excluded in the preparation of the ICs for CMAQ model simulations. Again, this
is because the Air Korea $NO_2$ mixing ratios are contaminated by several reactive nitrogen
species, so the data cannot be directly used in the assimilation procedures. In case of VOCs, a
limited number of datasets is available in South Korea for the DA. Improvements in the
prediction of ozone mixing ratios can be achieved when the $NO_2$ mixing ratios are corrected
and a sufficient number of VOCs data (possibly from satellite data in the future) is available.

**3.1.3. Statistical analysis**
In order to achieve better understanding of the performances of the DA RUN,
analyses of statistical variables such as index of agreement (IOA), Pearson's correlation





coefficient (R), root mean square error (RMSE), and mean bias (MB) were conducted using
observations from the Air Korea stations for $PM_{10}$, $PM_{2.5}$, CO, $SO_2$, and $O_3$ (see Fig. 11).
Definitions of the statistical variables are given in Appendix A.

After the applications of the ICs, both RMSE and MB became lower, while the

correlation coefficient became higher for the entire predictions. In addition, it was found that
the differences between the BASE RUN and the DA RUN tended to diminish as the
prediction time progressed. The results of the statistical analysis are listed in Table 1. The
results of the DA RUN were reasonably consistent with the observations for $PM_{10}$ (IOA =
0.60; R= 0.40; RMSE = 34.87; MB = -13.54) and $PM_{2.5}$ (IOA = 0.71; R= 0.53; RMSE = 17.
83; MB = -2.43), as compared to the BASE RUN for $PM_{10}$ (IOA = 0.51; R= 0.34; RMSE =
40.84; MB = -27.18) and $PM_{2.5}$ (IOA = 0.67; R= 0.51; RMSE = 19.24; MB = -9.9). In terms
of bias, an improvement was found for CO: MB = -0.036 for the DA RUN and MB = -0.27
for the BASE RUN. Regarding $O_3$ and $SO_2$, the DA RUN showed slightly better
performances than the BASE RUN.

Table 2 presents the results of the statistical analysis at 00:00 UTC when the DA was

conducted, with the results clearly showing how much closer the DA makes the CMAQ-
calculated chemical concentrations to the observed concentrations. Collectively, the DA
improved model accuracy by a large degree in terms of R, particularly for $PM_{10}$ (R:
0.3→0.75; slope: 0.17→0.66) and $O_3$ (R: 0.09→0.61; slope: 0.07→0.42). In addition, for all
species, MB and RMSE decreased significantly with the DA RUN as compared with the
BASE RUN.

**3.2. Sensitivity test of DA time-interval**



### 3.2.1. AOD


In this section, a sensitivity analysis was conducted with different implementation
time-intervals of the DA (i.e., 24, 6, and 3 hours) for AOD (refer to Fig. 12). As shown in Fig.
12, more frequent implementation of the DA is expected to make the predicted results closer
to the observations. Although the DA RUN with a shorter assimilation time-interval tends to
produce a better prediction, it is not always the most appropriate choice, since the shorter
assimilation time-interval results in increased computational cost. Therefore, an optimized
assimilation time-interval should be found to achieve the best performances from the given
DA system with the consideration of its own computational ability.

### 3.2.2. PM and gases


In addition, sensitivity analyses of the developed air quality prediction system to
multiple implementations of the DA with different time-intervals were also investigated for (a)
$PM_{10}$, (b) $PM_{2.5}$, (c) CO, (d) $SO_2$, and (e) $O_3$, shown in Fig. 13. Fig. 13 shows a soccer plot
analysis for BASE RUN (blue crosses) and DA RUNs with different DA time-intervals of 24
hours (OI; red circles), two hours (2-hr OI; black diamonds), and one hour (1-hr OI; dark-
green triangles). This set of testing was designed based on the fact that the performances are
expected to improve if the DAs are implemented multiple times prior to the actual predictions
at 00:00 UTC. Here, for the 2-hr OI run, the DA was implemented three times a day at 20:00,
22:00, and 00:00 UTC, while for the 1-hr OI run, the DA was implemented at 22:00, 23:00,
and 00:00 UTC. The performances of all of the chemical species excluding ozone improved,
as expected, with DA RUNs with more frequent and longer DA time-intervals (i.e., three-
times implementation with a 2-hr time-interval in our cases). In case of ozone, the best
performance was found for the air quality prediction system with the DA time-interval of 24-
hr.

Unsurprisingly, more frequent DAs prior to the actual prediction mode (i.e., before

00:00 UTC in our system) with a longer time-interval (such as 2-hr) will be computationally
costly. There will certainly be a "trade-off" between the precision of air quality prediction and
the computational cost. The system should be designed under the consideration of these two
factors.

**4. Summary and conclusions**

In this study, an operational air quality prediction system was developed by preparing

the ICs for CMAQ model simulations using GOCI AODs and ground-based observations of
$PM_{10}$, CO, ozone, and $SO_2$ during the period of the KORUS-AQ campaign (1 May – 12 June
2016) in South Korea. The major advantages of the developed air quality prediction system
are its comprehensiveness in predicting the ambient concentrations of both gaseous and
particulate species (including PM composition) as well as its powerfulness in terms of
computational cost.

The performances of the developed prediction system were evaluated using ground

in-situ observation data. The CMAQ model runs with the ICs (DA RUN) showed higher
consistency with the observations of almost all of the chemical species, including PM
composition (sulfate, nitrate, ammonium, OA, and EC) and atmospheric gases (CO, ozone,
and $SO_2$), than the CMAQ model runs without the ICs (BASE RUN). Particularly for CO, the
DA was able to remarkably improve the model performances, while the BASE RUN
significantly under-predicted the CO concentrations (predicting about one-third of the





observed values). In case of ozone, both the BASE RUN and DA RUN were in close
agreement with observations. More reliable predictions of ozone mixing ratios will be
achieved via the DA of the observed $NO_2$ mixing ratios and the corrections of model-
simulated mixing layer height (MLH). For $SO_2$, the performances of both the BASE RUN
and the DA RUN were somewhat poor. Regarding this issue, more accurate $SO_2$ emissions
are required to achieve better $SO_2$ predictions, and these can be estimated through inverse
modeling using satellite data (e.g., Lee et al., 2011). The adjustments of both ICs and
emissions may be able to improve the performances of the air quality prediction system, and
this will be examined in future studies.

Moreover, the developed air quality prediction system will be upgraded by using the

new observation data that will be retrieved after 2020 from the Geostationary Environment
Monitoring Spectrometer (GEMS) with a high spatial resolution of $7 \times 8$ $km^2$ as well as a
high temporal resolution of 1-hour over a large part of Asia. In addition, the current DA
technique of the OI with the Kalman filter can also be upgraded with the use of more
advanced DA methods such as variational techniques of 3DVAR and 4DVAR methods, as
well as with the ensemble Kalman filter (EnK) method. These research endeavors are
currently underway.

In conjunction with improving the air quality modeling system, artificial intelligence

(AI)-based air quality prediction systems are also currently being developed in several ways
(e.g., H. S. Kim et al., 2019). Both the CTM-based and AI-based air quality prediction
systems will be combined so as to ultimately enable more accurate air quality forecasts over
South Korea for Korean citizens. This is the ultimate goal of our research.



**Code and data availability.** WRF v3.8.1 (doi:10.5065/D6MK6B4K) and CMAQ v5.1
(doi:10.5281/zenodo.1079909) models are both open-source and publicly available. Source
codes for WRF and CMAQ can be downloaded at http://www2.mmm.ucar.edu/wrf/users/
downloads.html and https://github.com/USEPA/CMAQ, respectively. Data from the KORUS-
AQ field campaign can be downloaded from the KORUS-AQ data archive (http://www-
air.larc.nasa.gov/missions/korus-aq). Other data were acquired as follows. Ground-based
observation data were downloaded from the Air Korea website (http://www.airkorea.or.kr) for
South Korea and https://pm25.in for China. AERONET data were downloaded from
https://aeronet.gsfc.nasa.gov. All codes related with the air quality prediction system can be
obtained by contacting K. Lee (lkh1515@gmail.com).

**Author contributions.** KL developed the model code, performed the simulations, and
analyzed the results. CHS directed the experiments. JY contributed to shape the research and
analysis. SL, MP, HH, and SYP helped analyze the results. MC, JK, YK, JHW, and SWK
provided and analyzed data applied in the experiments. KL prepared the manuscript with
contributions from all co-authors.

**Acknowledgments**
This research was supported by the National Strategic Project-Fine particle of the National
Research Foundation of Korea (NRF) of the Ministry of Science and ICT (MSIT), the
Ministry of Environment (MOE), and the Ministry of Health and Welfare (MOHW) (NRF-
2017M3D8A1092022). This work was also funded by the MOE as "Public Technology



Program based on Environmental Policy (2017000160001)" and was supported by a grant
from the National Institute of Environment Research (NIER), funded by the MOE of the
Republic of Korea (NIER-2019-01-01-028). Specially thanks to the entire KORUS-AQ
science team for their considerable efforts in conducting the campaign.





**APPENDIX A: FORMULAS FOR STATISTICAL EVALUATION INDICES**
The formulas used to evaluate the performances of the operational air quality prediction
system are defined as follows.

$$\text{Index Of Agreement (IOA)} \ = \ 1 - \frac{\sum_1^n (M - O)^2}{\sum_1^n (|M - \overline{O}| + |O - \overline{O}|)^2} \quad \text{(A1)}$$

$$\text{Correlation Coefficient (R)} \ = \ \frac{1}{(n-1)} \sum_1^n \left( \left( \frac{O - \overline{O}}{\sigma_O} \right) \left( \frac{M - \overline{M}}{\sigma_m} \right) \right) \quad \text{(A2)}$$

$$\text{Root Mean Square Error (RMSE)} \ = \ \sqrt{\frac{\sum_1^n (M - O)^2}{n}} \quad \text{(A3)}$$

$$\text{Mean Bias (MB)} \ = \ \frac{1}{n} \sum_1^n (M - O) \quad \text{(A4)}$$

$$\text{Mean Normalized Bias (MNB)} \ = \ \frac{1}{n} \sum_1^n \left( \frac{M - O}{O} \right) \times 100\ \% \quad \text{(A5)}$$

$$\text{Mean Normalized Error (MNE)} \ = \ \frac{1}{n} \sum_1^n \left( \frac{|M - O|}{O} \right) \times 100\ \% \quad \text{(A6)}$$

$$\text{Mean Fractional Bias (MFB)} \ = \ \frac{1}{n} \sum_1^n \frac{(M - O)}{\left( \frac{M + O}{2} \right)} \times 100\ \% \quad \text{(A7)}$$





$$\text{Mean Fractional Error (MFE)} \; = \; \frac{1}{n}\sum_{1}^{n}\frac{|M-O|}{\left(\frac{M+O}{2}\right)} \times 100\;\% \quad (B8)$$

In Eqns. (A1) - (A8), M and O represent the model and observation data, respectively. N is
the number of data points and σ means the standard deviation. The overbars in the equations
indicate the arithmetic mean of the data. The units of RMSE and MB are the same as the unit
of data, while IOA and R are dimensionless statistical parameters.





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

Index products for land surface and climate modelling, Remote Sensing of Environment,
115(5), 1171–1187, doi:10.1016/j.rse.2011.01.001, 2011.

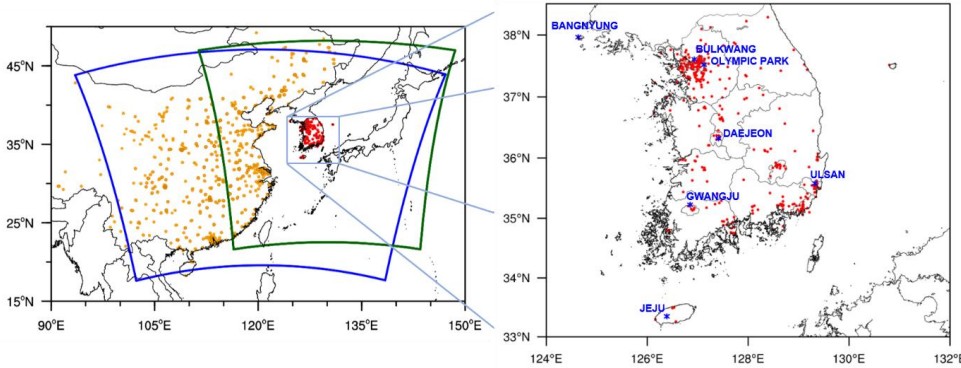


**Figure 1.** Domains of GOCI sensor (dark green line) and CMAQ model simulations (blue line). Red-colored dots denote the locations of Air Korea sites in South Korea. Orange-colored dots represent the locations of ground-based observation stations in China. Blue stars show the locations of seven super-sites in South Korea. During the KORUS-AQ campaign, observation data were obtained from 1514 stations in China as well as 264 Air Korea and seven super-site stations in South Korea.

762

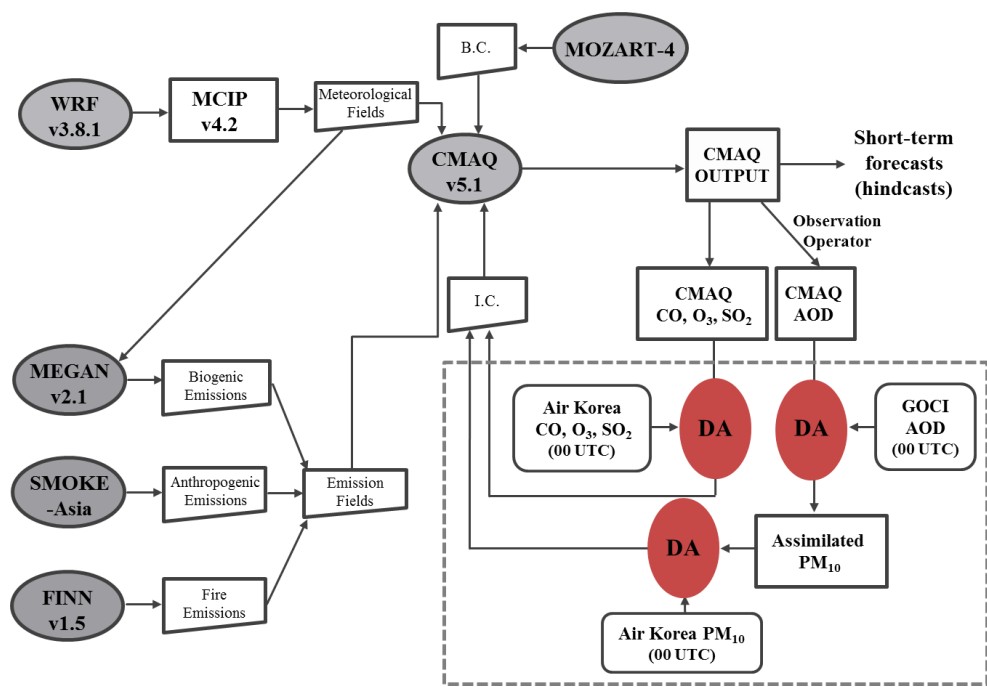

763

**Figure 2.** Schematic diagram of the Korean air quality prediction system developed in this study. The initial conditions (ICs) of the CMAQ model simulations are prepared by assimilating CMAQ outputs with satellite-retrieved and ground-measured observations. The data process for preparing the ICs is shown in the box with gray-dashed lines.

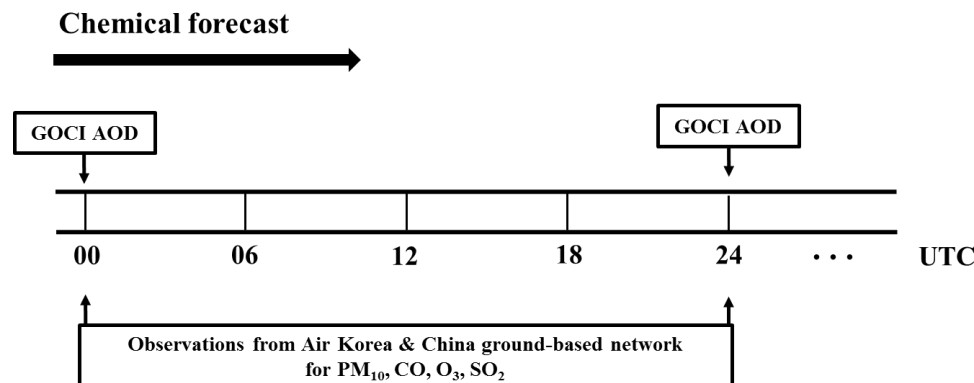

768

**Figure 3.** Schematic diagram of the Korean air quality prediction system for particulate matter (PM) and gas-phase pollutants. The data assimilation (DA) cycle is 24 hours for both PM and gas-phase pollutants such as CO, $O_3$, and $SO_2$. The DA of $NO_2$ is excluded in the current study, the reason for which is discussed in the text.



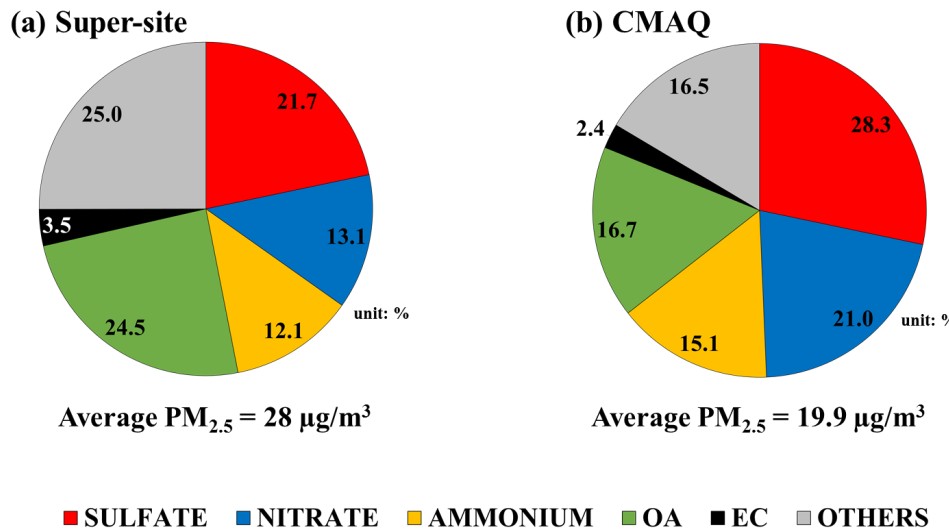

**Figure 4.** Average PM$_{2.5}$ composition (a) observed at the super-site stations and (b) simulated by the CMAQ model during the KORUS-AQ campaign. The averaged PM$_{2.5}$ measured from the super-sites and calculated from the CMAQ model simulations over the period of the KORUS-AQ campaign are 28 µg/m$^3$ and 19.9 µg/m$^3$, respectively. The mass of organic aerosols (OAs) was calculated by multiplying organic carbon mass by 1.6.

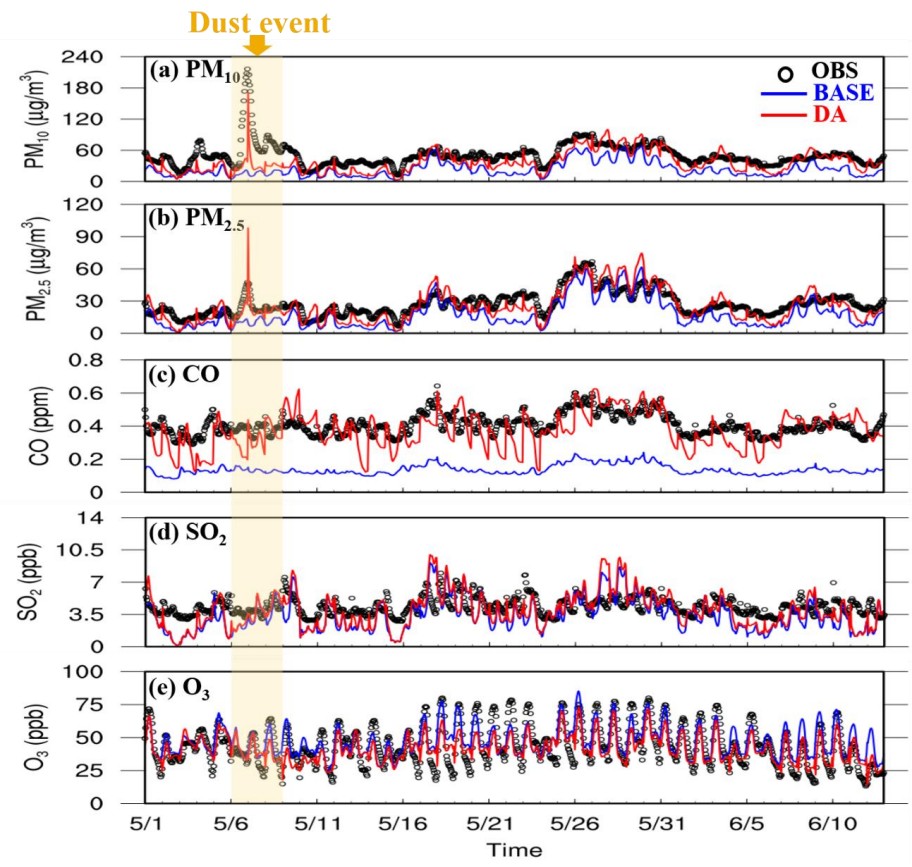

**Figure 5.** Time-series plots of hourly (a) $PM_{10}$, (b) $PM_{2.5}$, (c) CO, (d) $SO_2$, and (e) $O_3$ concentrations at 264 Air Korea stations. Black open circles (OBS) represent the observed concentrations. Blue and red lines show the results simulated from the BASE RUN and DA RUN over South Korea, respectively.



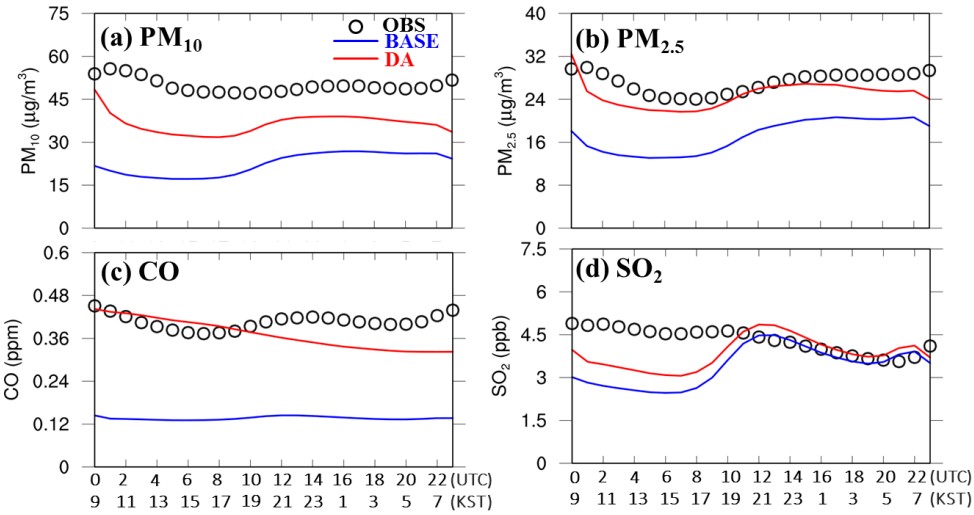

784

**Figure 6.** Aggregated average concentrations of (a) PM$_{10}$, (b) PM$_{2.5}$, (c) CO, and (d) SO$_2$ at 264 Air Korea stations over the KORUS-AQ campaign period. Open black circles denote the observations obtained from 264 Air Korea stations in South Korea. Blue and red lines represent the predicted concentrations from the BASE RUN and DA RUN, respectively. The DA was conducted at 00:00 UTC every day throughout the KORUS-AQ campaign period.



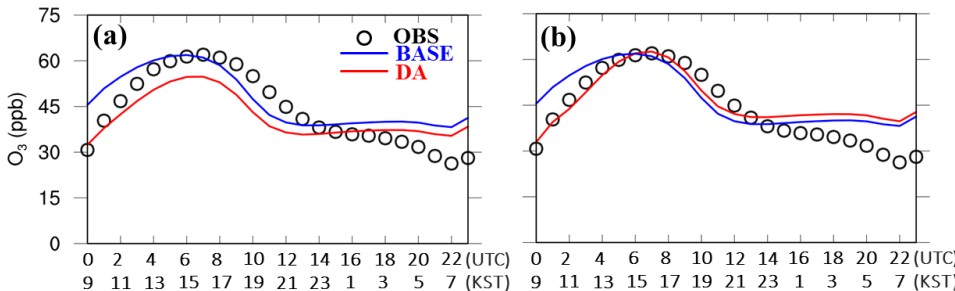

**Figure 7.** Comparison of CMAQ-simulated $O_3$ mixing ratios (BASE RUN with blue lines and DA RUN with red lines) with $O_3$ mixing ratios from Air Korea stations (open black circles). DA RUN was carried out by assimilating CMAQ outputs with Air Korea observations using (a) only $O_3$ mixing ratios and (b) both $O_3$ and $NO_2$ mixing ratios.

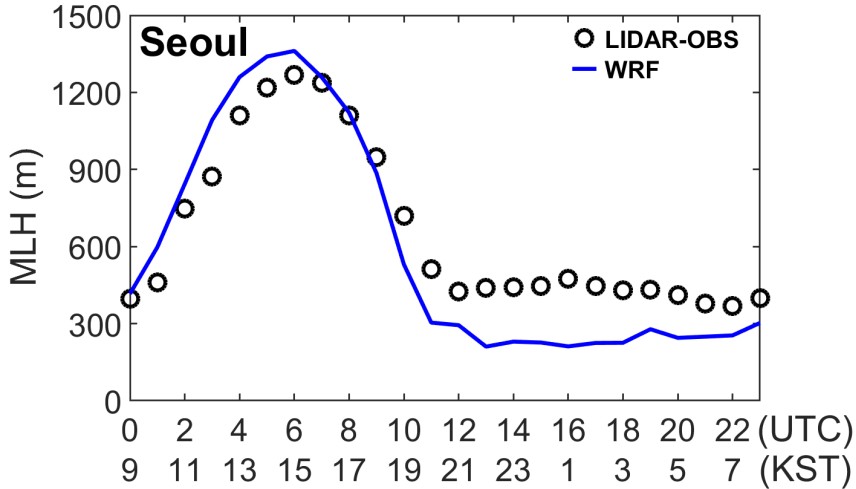

**Figure 8.** Comparison of WRF-simulated mixing layer height (MLH) (denoted by blue-dashed line) with lidar-measured MLH (denoted by open black circles) at Seoul National University (SNU) in Seoul. KST stands for Korean standard time.



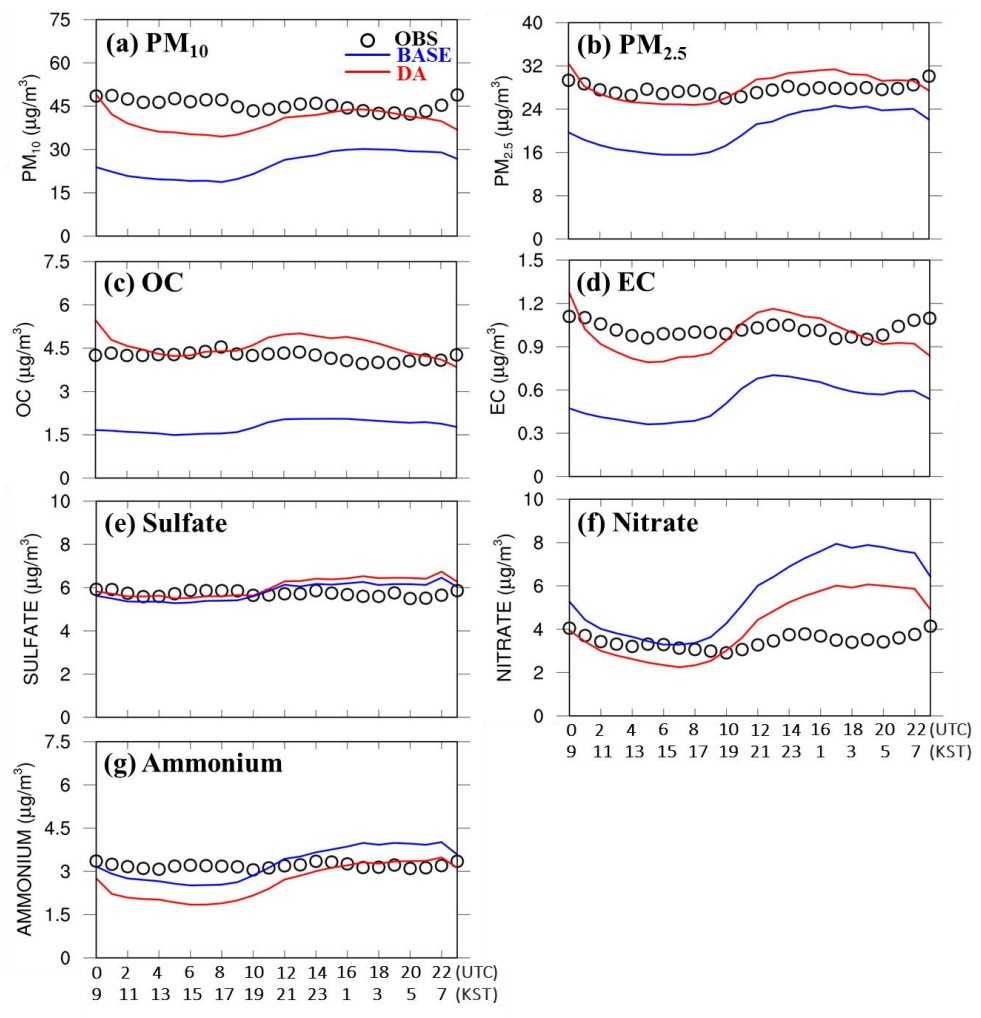

**Figure 9.** Aggregated average concentrations of (a) PM$_{10}$, (b) PM$_{2.5}$, (c) OC, (d) EC, (e) sulfate, (f) nitrate, and (g) ammonium as predicted by CMAQ model during the period of the KORUS-AQ campaign. The others are the same as those shown in Fig. 7, except for the fact that the observation data used here were obtained from the seven super-site stations in South Korea.

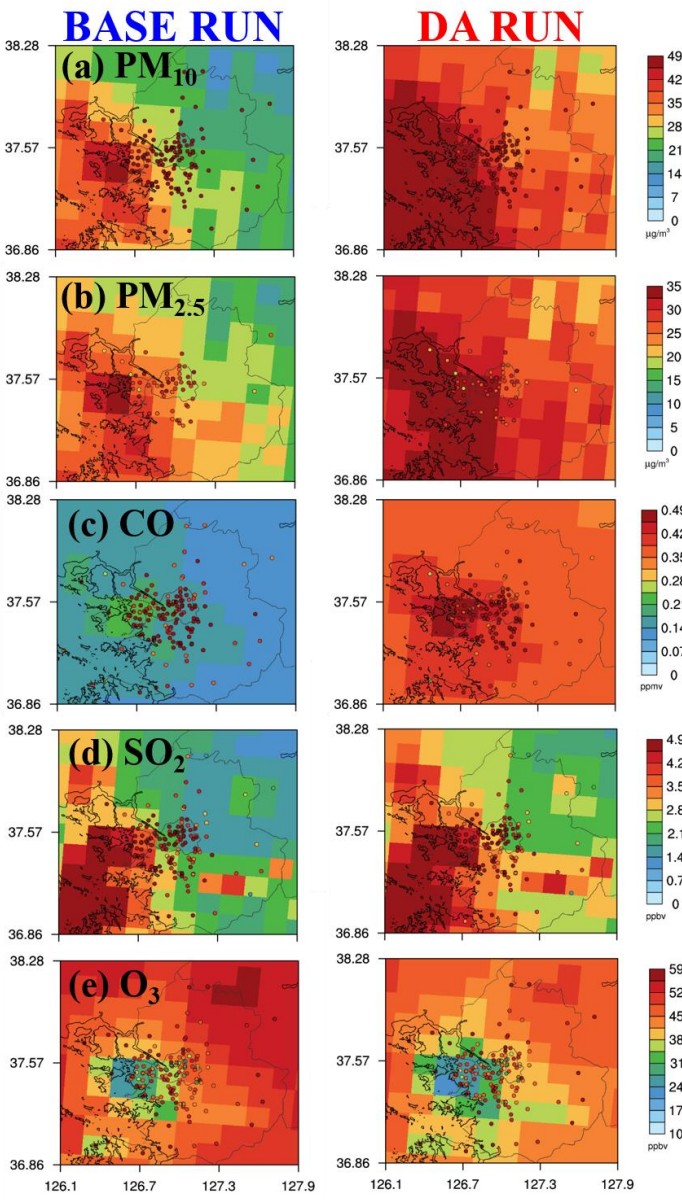

**Figure 10.** Spatial distributions of (a) PM$_{10}$, (b) PM$_{2.5}$, (c) CO, (d) SO$_2$, and (e) O$_3$ over Seoul Metropolitan Area (SMA). The concentrations were averaged over the entire period of the KORUS-AQ campaign. Colored circles represent the concentrations of the air pollutants observed at the Air Korea stations in the SMA.


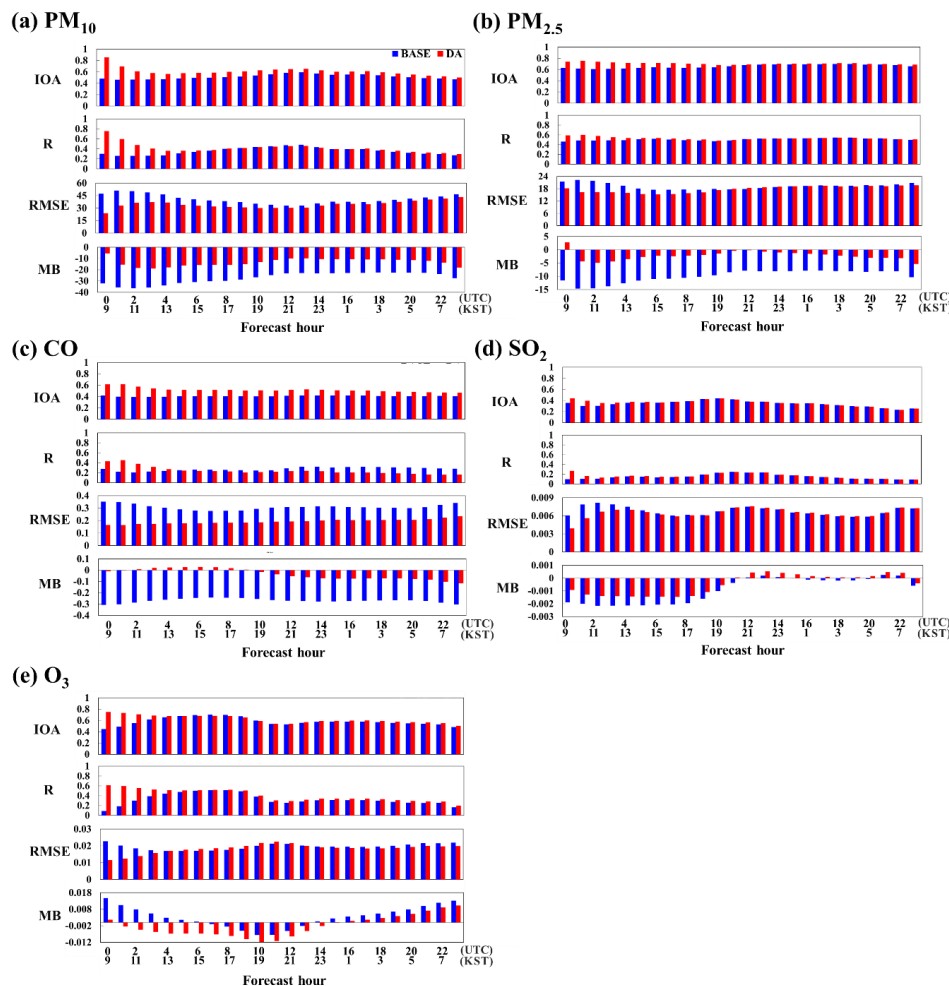

**Figure 11.** Time-series plots of four performance metrics (IOA, R, RMSE, and MB) for (a) $PM_{10}$, (b) $PM_{2.5}$, (c) CO, (d) $SO_2$, and (e) $O_3$ forecasts. The DA was conducted at 00:00 UTC. The units of RMSE and MB are $\mu g/m^3$ and ppmv for PM concentrations and for gaseous species, respectively. The definitions of the four performance metrics are shown in Appendix A.

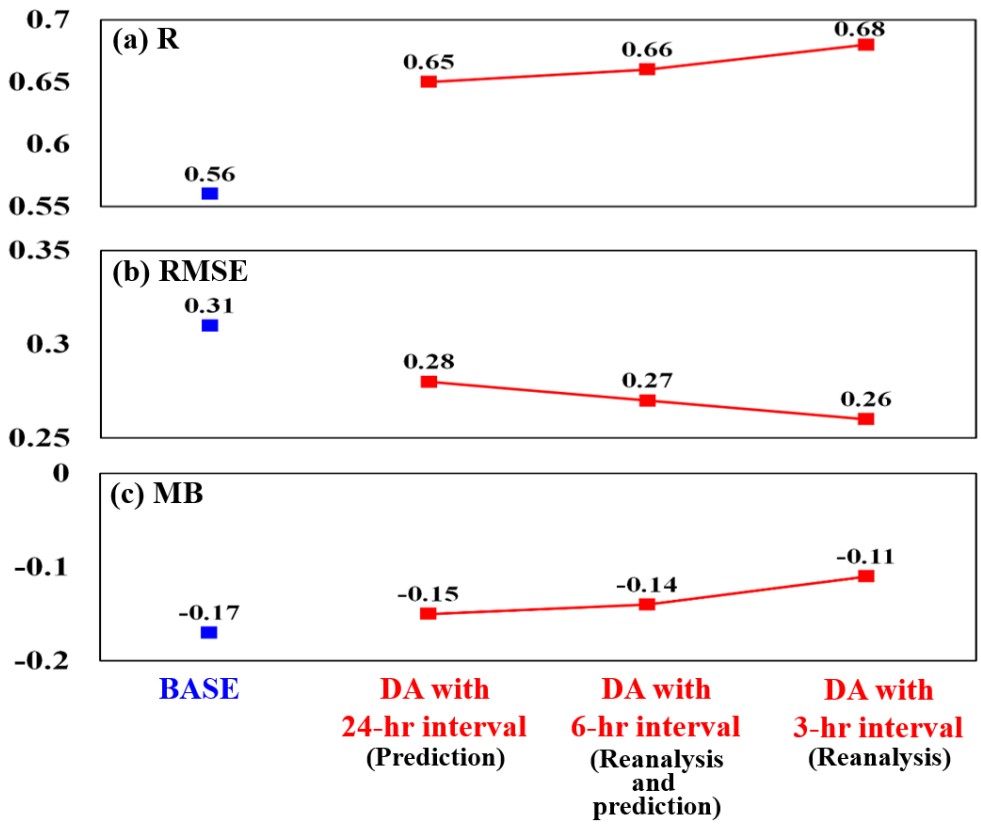

819

**Figure 12.** Variations of three performance metrics (R, RMSE, and MB) with time-intervals of data assimilations. For these tests, the GOCI AODs were used in the DA to update the initial conditions of the CMAQ model simulations. The results from the three CMAQ model simulations were compared with AERONET AODs ("ground truth"). The two blue squares represent the performances from the BASE RUNs and the red squares indicate the performances from the DA RUNs. The three experiments were carried out with the assimilation time-intervals of 24, 6, and 3 hours (hr), respectively. Here, the DA RUN with the 24-hr time-interval is referred to as "air quality prediction", and the DA RUNs with the 6-hr and 3-hr time-interval are referred to as "air quality reanalysis".

829

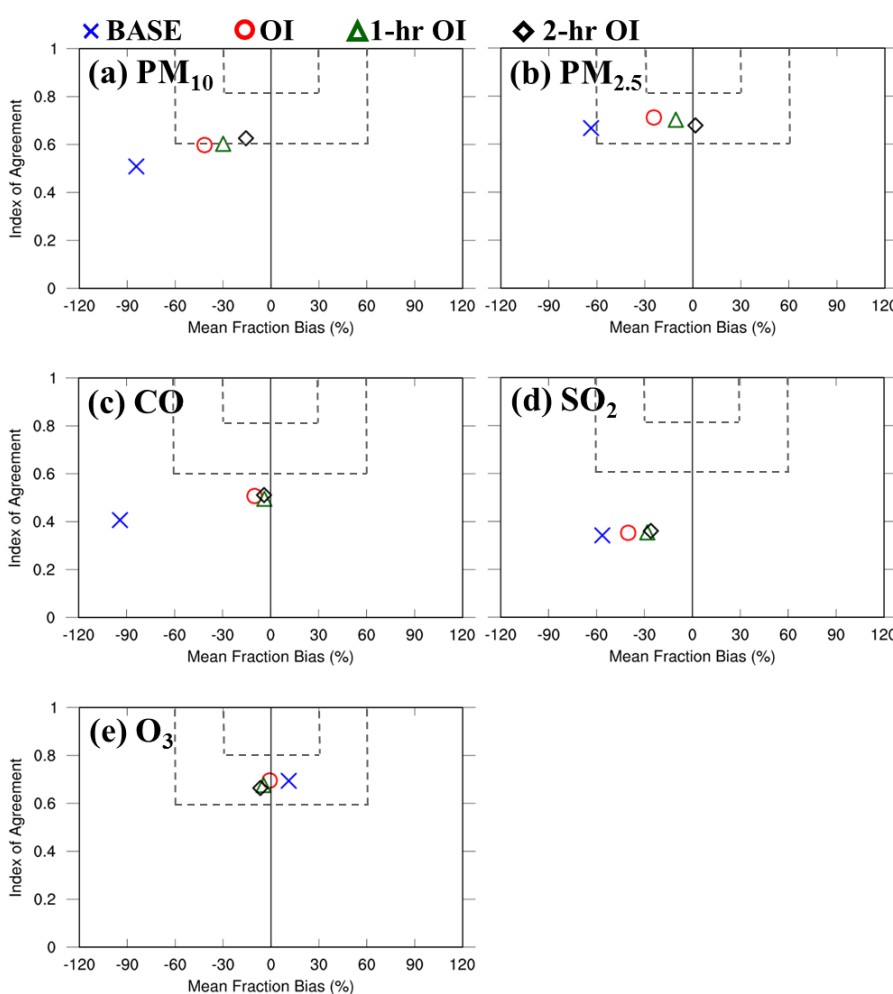

**Figure 13.** Soccer plot analyses for (a) PM$_{10}$, (b) PM$_{2.5}$, (c) CO, (d) SO$_2$, and (e) O$_3$. The CMAQ-predicted concentrations were compared with the Air Korea observations. Blue crosses, red circles, dark-green triangles, and black diamonds represent the performances calculated from the BASE RUN, the DA RUNs with the OI system, the 1-hour (hr) OI system, and the 2-hr OI system, respectively.





**Table 1.** Statistical metrics from BASE RUN and DA RUN with Air Korea observations over
the entire period of the KORUS-AQ campaign.

| | $PM_{10}$ | | $PM_{2.5}$ | | CO | | $SO_2$ | | $O_3$ | |
|---|---|---|---|---|---|---|---|---|---|---|
| | BASE RUN | DA RUN | BASE RUN | DA RUN | BASE RUN | DA RUN | BASE RUN | DA RUN | BASE RUN | DA RUN |
| N | 101852 | | 65383 | | 101764 | | 101764 | | 101836 | |
| IOA | 0.51 | 0.60 | 0.67 | 0.71 | 0.41 | 0.51 | 0.34 | 0.35 | 0.69 | 0.70 |
| R | 0.34 | 0.40 | 0.51 | 0.53 | 0.28 | 0.21 | 0.14 | 0.15 | 0.50 | 0.52 |
| RMSE | 40.8 | 34.87 | 19.2 | 17.83 | 0.31 | 0.19 | 0.0068 | 0.0066 | 0.020 | 0.02 |
| MB | -27.2 | -13.54 | -9.9 | -2.43 | -0.27 | -0.04 | -0.0009 | -0.0004 | 0.003 | -0.0024 |
| ME | 30.1 | 24.20 | 15.3 | 13.48 | 0.27 | 0.15 | 0.004 | 0.0034 | 0.015 | 0.015 |
| MNB | -50.0 | -18.17 | -30.1 | 5.32 | -62.0 | 3.14 | 3.1 | 17.77 | 48.0 | 30.22 |
| MNE | 60.7 | 52.35 | 62.6 | 62.77 | 62.9 | 40.67 | 93.1 | 93.56 | 70.2 | 61.34 |
| MFB | -84.3 | -41.61 | -63.6 | -24.41 | -94.1 | -10.00 | -56.4 | -40.20 | 11.1 | -0.82 |
| MFE | 91.1 | 62.32 | 81.6 | 60.01 | 94.9 | 39.49 | 91.4 | 82.91 | 40.7 | 40.64 |






**Table 2.** Statistical metrics from BASE RUN and DA RUN with Air Korea observations at
00:00 UTC when the DA was conducted during the KORUS-AQ campaign.

| | $PM_{10}$ | | $PM_{2.5}$ | | CO | | $SO_2$ | | $O_3$ | |
|---|---|---|---|---|---|---|---|---|---|---|
| | BASE RUN | DA RUN | BASE RUN | DA RUN | BASE RUN | DA RUN | BASE RUN | DA RUN | BASE RUN | DA RUN |
| N | 1057 | | 695 | | 1024 | | 1007 | | 1043 | |
| IOA | 0.48 | 0.86 | 0.63 | 0.74 | 0.41 | 0.62 | 0.36 | 0.44 | 0.45 | 0.75 |
| R | 0.30 | 0.75 | 0.46 | 0.59 | 0.28 | 0.43 | 0.097 | 0.27 | 0.09 | 0.61 |
| RMSE | 47.2 | 23.92 | 21.5 | 18.21 | 0.35 | 0.16 | 0.0061 | 0.0039 | 0.023 | 0.012 |
| MB | -32.2 | -5.46 | -11.5 | 2.80 | -0.31 | -0.01 | -0.0019 | -0.0009 | 0.015 | 0.002 |
| ME | 34.5 | 16.03 | 17.2 | 13.25 | 0.31 | 0.12 | 0.0039 | 0.0023 | 0.018 | 0.009 |
| MNB | -54.9 | -0.53 | -33.2 | 26.17 | -64.3 | 9.69 | -20.1 | 7.35 | 100.4 | 27.45 |
| MNE | 64.0 | 36.07 | 63.1 | 59.77 | 64.8 | 30.69 | 86.7 | 55.27 | 107.8 | 43.81 |
| MFB | -92.8 | -13.38 | -67.3 | 0.56 | -98.7 | 1.81 | -75.9 | -17.39 | 43.7 | 12.16 |
| MFE | 98.8 | 38.41 | 84.3 | 48.30 | 99.1 | 27.14 | 99.9 | 56.23 | 52.9 | 31.53 |
