# Peer review of "Development of Korean Air Quality Prediction System version 1 (KAQPS v1) with focuses on practical issues"

_Geoscientific Model Development, 2019_

## Referee Comment (RC1) · Anonymous Referee #1 · 11 Sep 2019

This is a straightforward manuscript describing a WRF-CMAQ system with data assimilation adjustment for its initial condition, and shows that the data assimilation improve the predictions in most cases. Here are some specified comments.

The data assimilation method mentioned mainly include AOD assimilation and surface measurement assimilation. Which one play a more important role for PM10 and PM2.5 adjustment? Do these adjustments have any conflict? Table 1 shows that CO's R and SO2's MNB become worse after data assimilation, why? Any discussion about it

[Figure]

Other minor issues: Line 276 (page 12), "Tang et al., 2017" can not be found in the reference. Line 282 (page 13), equation (7): the term "I" does not come with explanation in the text.

---

## Referee Comment (RC2) · Anonymous Referee #2 · 19 Sep 2019

General Comments: This paper discussed an application of offline WRF-CMAQ simulations and evaluation both with and without DA during the KORUS-AQ measurement campaign in Korea during May 1 – June 12, 2016. While the paper is well written, and the DA methodology is sound, there are some concerns on missing details and discussion throughout the paper (see Specific comments), rather expected model performance results, and using a such a case study/campaign application as an appropriate demonstration for the readiness of an "operational" air quality forecast system (below). While I recommend accepting the article for publication, these issues should

be addressed.

I think it is a bit misleading to call this application of WRF-CMAQ simulations a true "operational" air quality forecasting system, as it is currently only applied for a very short time period (May 1 – June 12) during a detailed measurement campaign. Such detailed observations are likely not available to demonstrate a continuous refresh of satellite/surface observations for DA in an operational system. To support this model system's use as "operational", what confidence is there that the statistical performance will scale to the remainder of the calendar year under different meteorological and chemical (i.e., emissions) environments? I think the "operational" focus of the paper should be dialed back in the paper, in place of a focus on the application of offline WRF-CMAQ chemical DA and evaluation during an intensive measurement campaign in Korea. Towards the end of the paper, it could be discussed of this WRF-CMAQ configuration could be further developed and more comprehensively tested to become an operational air quality forecasting system for Korea.

This issue is compounded by the somewhat expected results in the paper of improved model performance by assimilating data (compared to no DA), the impacts of a higher frequency rate of DA, and lack of testing the important "trade-off" that exists between increased precision and computational cost of DA. In fact, the authors explicitly state that this unsurprising result needs to be further tested, and that the true operational system should be designed under these considerations. These facts further substantiate my concern that it may be premature to consider this paper a description of an "operationally ready" model, but rather an application of testing DA in CMAQ during a relatively intensive KORUS campaign, which is a region that would certainly benefit from further development and testing of an operational air quality forecast model.

GMD Review Questions: 1. Does the paper address relevant scientific modelling questions within the scope of GMD? Does the paper present a model, advances in modelling science, or a modelling protocol that is suitable for addressing relevant scientific questions within the scope of EGU?

Yes.

2. Does the paper present novel concepts, ideas, tools, or data?

Yes.

3. Does the paper represent a sufficiently substantial advance in modelling science?

Yes, but some exceptions (see Specific Comments).

4. Are the methods and assumptions valid and clearly outlined?

Yes.

5. Are the results sufficient to support the interpretations and conclusions?'

Yes, with a few exceptions.

6. Is the description sufficiently complete and precise to allow their reproduction by fellow scientists (traceability of results)? In the case of model description papers, it should in theory be possible for an independent scientist to construct a model that, while not necessarily numerically identical, will produce scientifically equivalent results. Model development papers should be similarly reproducible. For MIP and benchmarking papers, it should be possible for the protocol to be precisely reproduced for an independent model. Descriptions of numerical advances should be precisely reproducible.

Yes.

7. Do the authors give proper credit to related work and clearly indicate their own new/original contribution?

Yes.

8. Does the title clearly reflect the contents of the paper? The model name and number should be included in papers that deal with only one model.

Not necessarily, as I have issues with this model being a true "operational" ready at

this point. See General and Specific Comments.

9. Does the abstract provide a concise and complete summary?

Yes.

10. Is the overall presentation well structured and clear?

Yes.

11. Is the language fluent and precise?

Yes.

12. Are mathematical formulae, symbols, abbreviations, and units correctly defined and used?

Yes.

13. Should any parts of the paper (text, formulae, figures, tables) be clarified, reduced, combined, or eliminated?

No.

14. Are the number and quality of references appropriate?

No. There are missing references.

15. Is the amount and quality of supplementary material appropriate? For model description papers, authors are strongly encouraged to submit supplementary material containing the model code and a user manual. For development, technical, and benchmarking papers, the submission of code to perform calculations described in the text is strongly encouraged.

Yes.

Specific Comments:

[Figure]

S1. Lines 88 – 93: Don't like the interchanging of air quality and chemical weather terminology here, as it is too similar to suggest that online chemical weather feedbacks are necessary in air quality models. I suggest revising to avoid any confusion.

S2. Lines 95 – 96: This is partially being overcome by new high spatial and temporal resolution satellite observations of composition (e.g., TEMPO, TROPOMI, GEMS, etc.). I think although they lack the longer term records, it should be mentioned that strides are being made at tackling these issues for air composition observations.

S3. Lines 98-101: What about the issues of coarse model grid scale for CTMs?

S4. Lines 142-155: This section is lacking information. 1) What are the dynamical/physical configurations for WRF (e.g., LSM, land use data, sfc layer, PBL, grid scale microphysics, convective cloud parameterization, etc.) ? 2) The met processor needs to be defined (i.e., MCIP) and explained for important derived variables from WRF to drive CMAQ. Many physical inconsistencies can arise between WRF-MCIP-CMAQ, and this is pivotal information in understanding the physical linkages between the upstream physics in WRF to drive CMAQ. 3) How exactly is OBSGRID applied in this model, and how does it relate to the WRF physical configurations? 4) Why was 15x15 km chosen, as opposed to commonly applied forecast models at 12x12 km? While much of this could be provided in supplemental/appendix to preserve brevity in the main text, it still needs to be included somewhere in the manuscript.

S5. Lines 157-177: This section is also lacking information. 1) What are the main chemical configurations for CMAQ (e.g., gas-phase chemistry, aerosol mechanism/size, dust/sea-salt, aqueous phase, dry/wet deposition, etc.)? 2) How does the CMAQ configurations interact and physically link with the upstream, driving physical configurations of WRF (comment S4)? 3) Clarification is needed on why MEGAN, rather than in-line BEIS, was used for biogenics in CMAQ. Is this based on literature for the Korean region?

S6. Lines 230 – 255: I think this is an important discussion, but please make it clear

how much this is a different formulation of AOD compared to the other pre-existing AOD calculations in CMAQv5.1 (e.g., the reconstruction method).

S7. Lines 311 – 316: I have some issue with assuming that $\Delta$PM2.5 exactly scales with $\Delta$PM10, because the inherent differences in some of the sources that make up the PM10 bias compared to PM2.5. In other words, if most of the $\Delta$PM10 is due to missing coarse mode aerosol emissions (e.g., dust etc.), we wouldn't expect this difference to have the same effect on $\Delta$PM2.5.

S8. Lines 391 – 392: Could this also be due to underpredicted NO2 with the DA run and not enough nighttime ozone titration? This perhaps could be better explored using Ox relationships and looking into different regions of the domain.

S9. Lines 397 – 398: It is concerning to draw such a conclusion based only on a single SNU lidar site comparison. Is this truly a widespread issue for nocturnal boundary layers in Korea? While this may indeed be common and well-defined previously using similar WRF physical options as in this study, there needs to be appropriate references here to provide support for your argument.

S10. Lines 406 – 408: This methodology is not clear. What set of MLH observations would be used for this effort? It certainly cannot be based on the single SNU lidar site. How would this be done "operationally" in the offline MCIP step between WRF and CMAQ? Bias correcting the MLH may lead to physical inconsistencies with other driving meteorological fields from WRF that were based on a particular set of physical configurations. Overall, this requires more thought likely, raises some concern, and should probably just be removed from the paper.

S11. Lines 412 - 413: CMAQ already has the "capability" to predict aerosol composition. Thus, it should be restated to say a "...a strong capability of our DA system is to improve predictions of CMAQ aerosol composition".

S12. Lines 416 – 424: These changes in model performance would be elucidated

if an addition column showing the absolute bias difference plot (colored in Red Blue shading) for the two runs compared to surface observations in Figure 10. This can be achieved by interpolating the closest model to the observations points.

S13. Lines 428 – 429: This is confusing, as it appears you are talking about an additional model simulation to the DA run. I thought that the adjusted NOx observations are used in the DA run, as discussed for the results in Lines 363-373 and comparing results from Figures 7a-b.

S14. Lines 525 – 529: Is the AI referring to its application to rapid refresh of emissions, chemical reaction/mechanism replacements, or something else? Is there a large body of research that shows AI can even "improve" air quality forecasts? I think the body of work shows that AI can be used speed up the gas chemistry in regional CTMs, while not suffering model performance degradation. Also, if the provided citation to Kim et al. (2019) is of no help, because it is not included in the reference list (see T6 correction below).

Technical Corrections:

T1. Line 120: Tang et al. (2017) is not found in the reference list.

T2. Line 145: Replace "dynamic" with "dynamical".

T3. Line 147: Replace "National Centers for Environmental Prediction Final Analysis data (NCEP FNL)" with "National Centers for Environmental Prediction (NCEP) Final (FNL) Operational Global Analysis data on 1°x 1° grids". Is 1°x 1° correct?

T4. Line 413. Replace "matters" with "matter"

T5. Line 501: Replace "ground" with "near-surface"

T6. Line 527: Kim et al. (2019) is not found in the reference list.
* * *
[Figure]

2019.

---

## Author Comment (AC1) · 12 Nov 2019

**Response to reviewer 2**

Authors appreciate reviewer's thoughtful comments and suggestions, which are greatly helpful for us to improve our manuscript. The manuscript has been revised to accommodate the reviewer's comments.

**General comment and response:**

**General Comments:** This paper discussed an application of offline WRF-CMAQ simulations and evaluation both with and without DA during the KORUS-AQ measurement campaign in Korea during May 1 – June 12, 2016. While the paper is well written, and the DA methodology is sound, there are some concerns on missing details and discussion throughout the paper (see Specific comments), rather expected model performance results, and using a such a case study/campaign application as an appropriate demonstration for the readiness of an "operational" air quality forecast system (below). While I recommend accepting the article for publication, these issues should be addressed.

I think it is a bit misleading to call this application of WRF-CMAQ simulations a true "operational" air quality forecasting system, as it is currently only applied for a very short time period (May 1 – June 12) during a detailed measurement campaign. Such detailed observations are likely not available to demonstrate a continuous refresh of satellite/surface observations for DA in an operational system. To support this model system's use as "operational", what confidence is there that the statistical performance will scale to the remainder of the calendar year under different meteorological and chemical (i.e., emissions) environments? I think the "operational" focus of the paper should be dialed back in the paper, in place of a focus on the application of offline WRF-CMAQ chemical DA and evaluation during an intensive measurement campaign in Korea. Towards the end of the paper, it could be discussed of this WRF-CMAQ configuration could be further developed and more comprehensively tested to become an operational air quality forecasting system for Korea.

This issue is compounded by the somewhat expected results in the paper of improved

model performance by assimilating data (compared to no DA), the impacts of a higher frequency rate of DA, and lack of testing the important "trade-off" that exists between increased precision and computational cost of DA. In fact, the authors explicitly state that this unsurprising result needs to be further tested, and that the true operational system should be designed under these considerations. These facts further substantiate my concern that it may be premature to consider this paper a description of an "operationally ready" model, but rather an application of testing DA in CMAQ during a relatively intensive KORUS campaign, which is a region that would certainly benefit from further development and testing of an operational air quality forecast model.

**Response:** Since we agreed that further researches are needed for true "operational" air quality prediction system, the word of "operational" has been removed from the original manuscript, and the title of the paper has also been modified to "Development of Korean Air Quality Prediction System version 1 (KAQPS v1) with focuses on practical issues" in order to avoid any confusion.

Actually, based on this work, the multiple-year tests are being currently conducted with the current sets of WRF-CMAQ-DA system. Once these tests are finished, we will revisit and report the issue of the "operational" air quality prediction system in South Korea, again.

**Specific comments and response:**

**Comment:** S1. Lines 88 – 93: Don't like the interchanging of air quality and chemical weather terminology here, as it is too similar to suggest that online chemical weather feedbacks are necessary in air quality models. I suggest revising to avoid any confusion.

**Response:** All of the terminology "chemical weather" has been replaced with "air quality" in the manuscript to avoid this confusion. Please, check out pp. 4:85 – 90.

**Comment:** S2. Lines 95 – 96: This is partially being overcome by new high spatial and temporal resolution satellite observations of composition (e.g., TEMPO, TROPOMI, GEMS, etc.). I think although they lack the longer term records, it should be mentioned that strides are being made at tackling these issues for air composition observations.

**Response:** We totally agree with referee's comment. A paragraph has been added to the revised paper for explaining the efforts to improve spatial and temporal coverage of satellite measurements. Please, refer to the added parts (pp. 5:93 – 101).

**Comment:** S3. Lines 98-101: What about the issues of coarse model grid scale for CTMs?

**Response:** Coarse-grid scale for CTM simulations can increase model uncertainties for several cases (Shrestha et al., 2009; Sirithian and Thepanondh, 2016), but not for all. After consideration of the matter, we have decided not to address this issue in detail, because it is beyond the scope of our manuscript.

**Comment:** S4. Lines 142-155: This section is lacking information.

1) What are the dynamical/physical configurations for WRF (e.g., LSM, land use data, sfc layer, PBL, grid scale microphysics, convective cloud parameterization, etc.) ?

**Response:** The dynamical and physical configurations for the WRF model simulations were added in the revised paper. Please, see pp. 7:151 – 157.

2) The met processor needs to be defined (i.e., MCIP) and explained for important derived variables from WRF to drive CMAQ. Many physical inconsistencies can arise between WRF-MCIPCMAQ, and this is pivotal information in understanding the physical linkages between the upstream physics in WRF to drive CMAQ.

**Response:** Information of MCIP has been added to the manuscript. Please, check out pp. 8:169 – 174.

3) How exactly is OBSGRID applied in this model, and how does it relate to the WRF physical configurations?

**Response:** The OBSGRID does not relate to the WRF physical configurations. This process was conducted to improve the accuracy of initial and boundary conditions for the WRF model simulation via data assimilation technique. A short sentence explaining OBSGRID has been added with more detailed information. Please, check this out at pp. 8:160 – 165.

4) Why was 15x15 km chosen, as opposed to commonly applied forecast models at 12x12 km? While much of this could be provided in supplemental/appendix to preserve brevity in the main text, it still needs to be included somewhere in the manuscript.

**Response:** Based on Lee et al. (2016)'s work, we chose 15 km by 15 km for CMAQ.

**Comment:** S5. Lines 157-177: This section is also lacking information.

1) What are the main chemical configurations for CMAQ (e.g., gas-phase chemistry, aerosol mechanism/size, dust/sea-salt, aqueous phase, dry/wet deposition, etc.)?

**Response:** The main chemical and physical configurations for the CMAQ model simulations were added into the revised manuscript. Please, refer to pp. 8:180 – pp. 9:189.

2) How does the CMAQ configurations interact and physically link with the upstream, driving physical configurations of WRF (comment S4)?

**Response:** Information on physical configurations of WRF and CMAQ model simulations has been added to the revised manuscript (refer to the responses to comments S4 and S5).

3) Clarification is needed on why MEGAN, rather than in-line BEIS, was used for biogenics in CMAQ. Is this based on literature for the Korean region?

**Response:** MEGAN was applied to the CMAQ model simulations in this study, because the MEGAN has been utilized in Korean modeling community and has also been widely used in many studies focusing on East Asia including South Korea (Kim et al., 2014; Kim et al., 2017; Lee et al., 2016; Park et al., 2014; Souri et al., 2017).

**Comment:** S6. Lines 230 – 255: I think this is an important discussion, but please make it clear how much this is a different formulation of AOD compared to the other pre-existing AOD calculations in CMAQv5.1 (e.g., the reconstruction method).

**Response:** A paragraph has been added in the revised paper. Please, see pp. 13:277 – 286.

**Comment:** S7. Lines 311 – 316: I have some issue with assuming that $\Delta PM_{2.5}$ exactly scales with $\Delta PM_{10}$, because the inherent differences in some of the sources that make up the $PM_{10}$ bias compared to $PM_{2.5}$. In other words, if most of the $\Delta PM_{10}$ is due to missing coarse mode aerosol emissions (e.g., dust etc.), we wouldn't expect this difference to have the same effect on $\Delta PM_{2.5}$.

**Response:** Yes, it may be a correct point! Our problem has been that the missing sources of the coarse-mode discrepancy (i.e., $\Delta PM_{2.5-10}$) have not been identified in South Korea. We are thinking that the uncertainty in the fugitive dust emissions from construction sites, road sites, cattle-raising areas, dry mud fields, etc may take significant responsibility for the $\Delta PM_{2.5-10}$ in South Korea. However, the amounts of such emissions from individual source have been difficult to quantify. In addition to a method we used in this study, we have to add the above amounts in the future study. This may be the reason why Fig. 9 shows larger differences in the $PM_{10}$ predictions than in the $PM_{2.5}$ predictions.

**Comment:** S8. Lines 391 – 392: Could this also be due to underpredicted $NO_2$ with the DA run and not enough nighttime ozone titration? This perhaps could be better explored using Ox relationships and looking into different regions of the domain.

**Response:** A following sentence has been added into the revised manuscript. Please, see pp. 19:424 – 428.

**Comment:** S9. Lines 397 – 398: It is concerning to draw such a conclusion based only on a single SNU lidar site comparison. Is this truly a widespread issue for nocturnal boundary layers in Korea? While this may indeed be common and well-defined previously using similar WRF physical options as in this study, there needs to be appropriate references here to provide support for your argument.

**Response:** Although we showed only one site example here, this nocturnal MLH problem has been commonly found in South Korea. Korean modeling community have also been well aware of this problem for a long time (Kang et al., 2016; Nam et al., 2016). A sentence in Lines 397-398 of the original manuscript has been modified, because a more intensive comparison study between lidar-retrieved and model-simulated MLH is necessary. Please, see pp. 20:435 – 437.

**Comment:** S10. Lines 406 – 408: This methodology is not clear. What set of MLH observations would be used for this effort? It certainly cannot be based on the single SNU lidar site. How would this be done "operationally" in the offline MCIP step between WRF and CMAQ? Bias correcting the MLH may lead to physical inconsistencies with other driving meteorological fields from WRF that were based on a particular set of physical configurations. Overall, this requires more thought likely, raises some concern, and should probably just be removed from the paper.

**Response:** Following reviewer's suggestion, the related sentences of Lines 406-408 have been removed from the manuscript since we agree that careful consideration of

the MLH bias correction is needed. We are thinking that the bias correction will not lead to physical inconsistency in the "off-line" mode modeling, but it can create a problem in the "on-line (two-way)" modeling.

**Comment:** S11. Lines 412 - 413: CMAQ already has the "capability" to predict aerosol composition. Thus, it should be restated to say a ": : :a strong capability of our DA system is to improve predictions of CMAQ aerosol composition".

**Response:** We modified the corresponding sentence into "a strong capability of our DA system is to improve predictions of CMAQ aerosol composition". Please, check this out at pp. 20:441 – 442.

**Comment:** S12. Lines 416 – 424: These changes in model performance would be elucidated if an addition column showing the absolute bias difference plot (colored in Red Blue shading) for the two runs compared to surface observations in Figure 10. This can be achieved by interpolating the closest model to the observations points.

**Response:** Yes, it is a good idea! Following reviewer's suggestion, bias difference plot has been added into Fig. 10, and the caption of the figure has also been changed. Please, check out the modified Fig. 10 at pp. 45 in the revised manuscript.

[Figure]

**Figure 10.** Spatial distributions (first and second columns) and bias (third and fourth columns) of (a) PM$_{10}$, (b) PM$_{2.5}$, (c) CO, (d) SO$_2$, and (e) O$_3$ over Seoul Metropolitan Area (SMA) for the entire period of the KORUS-AQ campaign. Colored circles of first and second columns represent the concentrations of the air pollutants observed at the Air Korea stations in the SMA.

**Comment:** S13. Lines 428 – 429: This is confusing, as it appears you are talking about an additional model simulation to the DA run. I thought that the adjusted NOx observations are used in the DA run, as discussed for the results in Lines 363-373 and comparing results from Figures 7a-b.

**Response:** As addressed in Lines 428-429 of the original manuscript, Air Korea observations for $NO_2$ were not applied in a 100 % accurate way in the current version of Korean air quality prediction system. It is because $NO_2$ mixing ratios measured at the Air Korea sites are contaminated by other nitrogen gases due to "$NO_2$ measurement artifacts" as discussed in Lines 374-387 (original manuscript). In Fig. 7(a), the results of a DA RUN by assimilating CMAQ outputs with Air Korea-observed $O_3$ mixing ratios are shown. Fig. 7(b) depicts the results of the test run by assimilating CMAQ outputs with Air Korea-observed both $O_3$ and $NO_2$ mixing ratios. But, here we used "$0.8 \times NO_2$ mixing ratios", following information given by Jung et al. (2017). We believe this work is not 100 % perfect! That's why we call the DA RUN a preliminary DA RUN. In the future, we attempt to correct these artifacts of $NO_2$ mixing ratios. Then, we will revisit the impacts of the $NO_2$ assimilation on $O_3$ mixing ratios. To avoid confusion regarding this issue, the corresponding sentence was modified in the revised manuscript (please, see pp. 18:394 – 396).

**Comment:** S14. Lines 525 – 529: Is the AI referring to its application to rapid refresh of emissions, chemical reaction/mechanism replacements, or something else? Is there a large body of research that shows AI can even "improve" air quality forecasts? I think the body of work shows that AI can be used speed up the gas chemistry in regional CTMs, while not suffering model performance degradation. Also, if the provided citation to Kim et al. (2019) is of no help, because it is not included in the reference list (see T6 correction below).

**Response:** Kim et al. (2019) recently published a paper in ACP that employed a deep recurrent neural network system based on long short-term memory (LSTM) model for daily $PM_{10}$ and $PM_{2.5}$ predictions in South Korea. The prediction system was optimized by iterative model trainings with the inputs of ground-based observations for $PM_{10}$, $PM_{2.5}$, and the observed meteorological variables including wind speed, wind direction, relative humidity, precipitation, etc. Their AI-based prediction system showed better performances than the CMAQ model simulations. However, the current AI system works only for points where ground-based observations are made. Therefore, we

expect that a combination of the AI system with the currently developed air quality prediction system can produce a more accurate air quality forecast over South Korea. Regarding this issue, please refer to pp. 25:550 – 559.

**Technical corrections and response:**

**Comment:** T1. Line 120: Tang et al. (2017) is not found in the reference list.

**Response:** "Tang et al., 2017" has been added into the references. Please, check this out at pp. 34:830 – 834.

**Comment:** T2. Line 145: Replace "dynamic" with "dynamical".

**Response:** "dynamic" has been replaced by "dynamical". Please, check this out at pp. 7:149.

**Comment:** T3. Line 147: Replace "National Centers for Environmental Prediction Final Analysis data (NCEP FNL)" with "National Centers for Environmental Prediction (NCEP) Final (FNL) Operational Global Analysis data on 1° × 1° grids". Is 1° × 1° correct?

**Response:** Yes, 1° × 1° is correct. "National Centers for Environmental Prediction Final Analysis data (NCEP FNL)" has been replaced with "National Centers for Environmental Prediction (NCEP) Final (FNL) Operational Global Analysis data on 1° x 1° grids". Please, see this out pp. 7:157 – pp. 8:159.

**Comment:** T4. Line 413. Replace "matters" with "matter"

**Response:** "matters" has been deleted in the manuscript, following the referee's advice (S11) . Thank you!

**Comment:** T5. Line 501: Replace "ground" with "near-surface"

**Response:** "ground" has been replaced with "near-surface". Please, check this out at pp. 24:526 – 527.

**Comment:** T6. Line 527: Kim et al. (2019) is not found in the reference list.

**Response:** "Kim et al. (2019)" has been added to the references. Please, check this out at pp. 32:752 – 755.

**References**

Kang, M., Lim, Y.-K., Cho, C., Kim, K. R., Park, J. S. and Kim, B.-J.: Accuracy Assessment of Planetary Boundary Layer Height for the WRF Model Using Temporal High Resolution Radio-sonde Observations, , 26, doi:10.14191/ATMOS.2016.26.4.673, 2016.

Kim, H. C., Kim, S., Kim, B.-U., Jin, C.-S., Hong, S., Park, R., Son, S.-W., Bae, C., Bae, M., Song, C.-K. and Stein, A.: Recent increase of surface particulate matter concentrations in the Seoul Metropolitan Area, Korea, Scientific Reports, 7(1), 4710, doi:10.1038/s41598-017-05092-8, 2017.

Kim, H. S., Park, I., Song, C. H., Lee, K., Yun, J. W., Kim, H. K., Jeon, M., Lee, J. and Han, K. M.: Development of a daily PM10 and PM2.5 prediction system using a deep long short-term memory neural network model, Atmos. Chem. Phys., 19(20), 12935–12951, doi:10.5194/acp-19-12935-2019, 2019.

Kim, H.-K., Woo, J.-H., Park, R. S., Song, C. H., Kim, J.-H., Ban, S.-J. and Park, J.-H.: Impacts of different plant functional types on ambient ozone predictions in the Seoul Metropolitan Areas (SMAs), Korea, Atmospheric Chemistry and Physics, 14(14), 7461–7484, doi:10.5194/acp-14-7461-2014, 2014.

Lee, S., Song, C. H., Park, R. S., Park, M. E., Han, K. M., Kim, J., Choi, M., Ghim, Y. S. and Woo, J.-H.: GIST-PM-Asia v1: development of a numerical system to improve particulate matter forecasts in South Korea using geostationary satellite-retrieved aerosol optical data over Northeast Asia, Geosci. Model Dev., 9(1), 17–39, doi:10.5194/gmd-9-17-2016, 2016.

Nam, H.-G., Choi, W., Kim, Y.-J., Shim, J.-K., Cho, B.-C. and Kim, B.-G.: Estimate and Analysis of Planetary Boundary Layer Height (PBLH) using a Mobile Lidar Vehicle system, , 32, doi:10.7780/KJRS.2016.32.3.9, 2016.

Park, M. E., Song, C. H., Park, R. S., Lee, J., Kim, J., Lee, S., Woo, J.-H., Carmichael, G. R., Eck, T. F., Holben, B. N., Lee, S.-S., Song, C. K. and Hong, Y. D.: New approach to monitor transboundary particulate pollution over Northeast Asia, Atmos. Chem. Phys., 14(2), 659–674, doi:10.5194/acp-14-659-2014, 2014.

Shrestha, K. L., Kondo, A., Kaga, A. and Inoue, Y.: High-resolution modeling and evaluation of ozone air quality of Osaka using MM5-CMAQ system, Journal of Environmental Sciences, 21(6), 782–789, doi:https://doi.org/10.1016/S1001-0742(08)62341-4, 2009.

Sirithian, D. and Thepanondh, S.: Influence of Grid Resolution in Modeling of Air Pollution from Open Burning, Atmosphere, 7(7), doi:10.3390/atmos7070093, 2016.

Souri, A. H., Choi, Y., Jeon, W., Woo, J.-H., Zhang, Q. and Kurokawa, J.: Remote sensing evidence of decadal changes in major tropospheric ozone precursors

over East Asia, Journal of Geophysical Research: Atmospheres, 122(4), 2474–2492, doi:10.1002/2016JD025663, 2017.

Tang, Y., Pagowski, M., Chai, T., Pan, L., Lee, P., Baker, B., Kumar, R., Delle Monache, L., Tong, D. and Kim, H.-C.: A case study of aerosol data assimilation with the Community Multi-scale Air Quality Model over the contiguous United States using 3D-Var and optimal interpolation methods, Geosci. Model Dev., 10(12), 4743–4758, doi:10.5194/gmd-10-4743-2017, 2017.

---

## Author Comment (AC2) · 12 Nov 2019

**Response to reviewer 1**

Authors appreciate reviewer's thoughtful comments and suggestions, which are greatly helpful for us to improve our manuscript. The manuscript has been revised to accommodate the reviewer's comments.

**General comment:**

This is a straightforward manuscript describing a WRF-CMAQ system with data assimilation adjustment for its initial condition, and shows that the data assimilation improve the predictions in most cases. Here are some specified comments.

**Comments and response:**

**Comment:** The data assimilation method mentioned mainly include AOD assimilation and surface measurement assimilation. Which one play a more important role for $PM_{10}$ and $PM_{2.5}$ adjustment? Do these adjustments have any conflict?

**Response:** To clarify the impacts of the AOD and PM adjustments, assimilation was conducted in the following procedure. First, CMAQ simulated-AODs were assimilated with GOCI-retrieved AODs, and the assimilated AODs were then allocated into the PM concentrations, based on model-simulated concentrations. After that, the allocated PM was again assimilated using ground-based PM observations. Therefore, the assimilation using the surface measurements played a more important role in the entire PM adjustments. The reasons why the GOCI AODs were applied prior to using ground-based observations are two-fold: (1) GOCI slightly underestimates AODs compared to the AODs measured by the AERONET, which possibly leads to underestimated PM adjustments and (2) the allocation of AOD into PM concentrations based on model values has uncertainties. Despite these two reasons, we found that GOCI AODs are still useful for data assimilation (DA), because satellites provide meaningful information especially over the ocean areas where no surface-based observations are available.

**Comment:** Table 1 shows that CO's R and $SO_2$'s MNB become worse after data assimilation, why?

**Response:** As listed in Table 1 (in this response), the model-calculated CO concentrations (BASE RUN) are by far lower than those observed by in-situ measurements. After conducting DA at 00:00 UTC, the CMAQ-simulated CO concentrations became closer to observations. Up to 6 hours, the DA RUN showed a better performance (R=0.56; MB=0.017) compared to the BASE RUN (R=0.40; MB=-0.27). However, the differences between the BASE RUN and the DA RUN were diminished as the prediction progressed, because model tends to go back to its original state. Because of this tendency, the scatter plot of the DA RUN became more widespread, i.e., smaller correlation coefficient, than those of the BASE RUN for 0 – 23 hours predictions.

In case of $SO_2$ (see Table 2 in this response), the DA RUN showed a better performance for 0 – 6 hours and 0 – 23 hours predictions compared with the BASE RUN in terms of R, IOA, RMSE, and MB. Unlike these statistical variables, MNB, a relative difference normalized by observations, was decreased in the DA RUN for 0 – 6 hours predictions and increased for 0 – 23 hours predictions. Figure 11(d) shows the discrepancy of MB between daytime and nighttime. The model-simulated $SO_2$ concentrations of both the BASE RUN and the DA RUN were much smaller than observations during the daytime, and became similar (BASE RUN) or larger (DA RUN) compared to observations during the nighttime. These over-predicted nocturnal $SO_2$ concentrations of the DA RUN lead to large positive MNB values. This can also be explained by the underestimated nocturnal mixing layer height (MLH) shown in Fig. 8. For further investigation, we are collecting and analyzing more lidar data available over South Korea. In the future, a further comparison study will be carried out using those lidar-measured MLH over South Korea.

**Table 1.** Statistical metrics for CO from BASE RUN and DA RUN with Air Korea observations at 00:00 UTC when the DA was conducted, 0 – 6 hr predictions after DA, and 0 – 23 hr predictions over the entire period of the KORUS-AQ campaign.

| CO | At DA time (00 UTC) | | 0 - 6 hr prediction | | 0 - 23 hr prediction | |
|---|---|---|---|---|---|---|
| | BASE RUN | DA RUN | BASE RUN | DA RUN | BASE RUN | DA RUN |
| N | 1024 | | 27268 | | 101764 | |
| IOA | 0.41 | 0.62 | 0.24 | 0.33 | 0.41 | 0.51 |
| R | 0.28 | 0.43 | 0.40 | 0.56 | 0.28 | 0.21 |
| RMSE | 0.35 | 0.16 | 0.31 | 0.17 | 0.31 | 0.19 |
| MB | -0.31 | -0.01 | -0.27 | 0.017 | -0.27 | -0.04 |
| MNB | -64.3 | 9.69 | -62.52 | 17.11 | -62.0 | 3.14 |

**Table 2.** Same as Table 1, except for $SO_2$.

| $SO_2$ | At DA time (00 UTC) | | 0 - 6 hr prediction | | 0 - 23 hr prediction | |
|---|---|---|---|---|---|---|
| | BASE RUN | DA RUN | BASE RUN | DA RUN | BASE RUN | DA RUN |
| N | 1007 | | 27258 | | 101764 | |
| IOA | 0.36 | 0.44 | 0.34 | 0.37 | 0.34 | 0.35 |
| R | 0.097 | 0.27 | 0.13 | 0.15 | 0.14 | 0.15 |
| RMSE | 0.0061 | 0.0039 | 0.0074 | 0.0065 | 0.0068 | 0.0066 |
| MB | -0.0019 | -0.0009 | -0.0021 | -0.0014 | -0.0009 | -0.0004 |
| MNB | -20.1 | 7.35 | -29.87 | -7.54 | 3.1 | 17.77 |

**Comment:** Line 276 (page 12), "Tang et al., 2017" cannot be found in the reference.

**Response:** "Tang et al., 2017" has been included to the references. Please, see pp. 34:830 – 834.

**Comment:** Line 282 (page 13), equation (7): the term "I" does not come with explanation in the text.

**Response:** The sentence of "I denotes the unit matrix" has been added to the manuscript. Please, see pp. 14:320 – pp. 15:321.

---

## Author Response (AR2)

**Cover letter**

**12 December, 2019**

Dr. Havala Pye

Topical editor of Geoscientific Model Development (GMD)

National Exposure Research Laboratory

U.S. Environmental Protection Agency

United States

**Attn:** Dr. Havala Pye (Topical editor)

**Re:** Manuscript, gmd-2019-169

Dear Dr. Havala Pye:

Thank you for your consideration of this manuscript entitled "Development of Korean Air Quality Prediction System version 1 (KAQPS v1) with focuses on practical issues". Since this research has been supported by the Korean Ministry of Science and ICT (MSIT), the Ministry of Environment (MOE), and the Ministry of Health and Welfare (MOHW), all the research results are, in principle, owned by the South Korean government. Therefore, the release of the source should be approved by the government. The code-sharing policy was not decided yet. In addition, in the future articles, we will use the Word difference tool to track the changes in the manuscript for responses to reviewers' comments as the editor recommended.

Again, thank you for your consideration and handling of this manuscript. We look forward to hearing from you regarding the final decision of this paper.

Sincerely,

Chul H. Song, Ph.D.

School of Earth Sciences and Environmental Engineering

Gwangju Institute of Science and Technology (GIST)

123 Cheomdan-gwagiro (Oryong-dong), Buk-gu, Gwangju 61005, Republic of Korea

Tel.: +82-62-715-3276

E-mail: chsong@gist.ac.kr

---

## Author Response (AR3)

**Response to Topical Editor**

Authors appreciate the Topical Editor's thoughtful comments, which are helpful to further improve the manuscript. The manuscript has been revised to accommodate the Topical Editor's comments. The modified parts are italicized in the replies.

**Comment and response:**

**Comment:** In the code/data availability statement, please provide a detailed licensing statement (in agreement with the MSIT if required) regarding code distribution. Since the code is not being publicly archived, GMD requires the restrictions be clearly stated.

**Response:** We have decided to provide the KAQPSv1 code publicly. The "Code and data availability" session related to the KAQPSv1 code has been modified as follows (please, see pp. 25:569 – pp. 26:570).

All codes related to the air quality prediction system can be obtained by contacting K. Lee (lkh1515@gmail.com).

→ *The KAQPS v1 code can be obtained by contacting K. Lee (lkh1515@gmail.com) or from https://github.com/AIR-Codes/KAQPSv1.*

---

## Author Response (AR4)

**Response to Topical Editor**

Authors appreciate the Topical Editor's thoughtful comments, which are helpful to further improve the manuscript. The manuscript has been revised to accommodate the Topical Editor's comments. The modified parts are italicized in the replies.

**Comment and response:**

**Comment:** Thank you for providing the github code. Could a persistent identifier (doi) be provided as well to comply with the best practice level? This is in case the github code changes in the future. Some authors have found zenodo, which interfaces with github, and reasonable way to obtain a doi, but other methods are available as well.

**Response:** According to the Topical Editor's advice, the doi number has been added to the manuscript using zenodo as follows (please, see pp. 25:569 – pp. 26:571).

The KAQPS v1 code can be obtained by contacting K. Lee (lkh1515@gmail.com) or from https://github.com/AIR-Codes/KAQPSv1.

→ *The KAQPS v1 (doi:10.5281/zenodo.3659551) code can be obtained by contacting K. Lee (lkh1515@gmail.com) or from https://github.com/AIR-Codes/KAQPSv1.*